# Reprogrammed CRISPR-Cas13b suppresses SARS-CoV-2 replication and circumvents its mutational escape through mismatch tolerance

Mohamed Fareh [1,2✉], Wei Zhao[3], Wenxin Hu [1,2], Joshua M. L. Casan [1,2], Amit Kumar[1,2], Jori Symons[3], Jennifer M. Zerbato [3], Danielle Fong [3], Ilia Voskoboinik[1,2], Paul G. Ekert [1,2,4,5], Rajeev Rudraraju[3,6,7], Damian F. J. Purcell[7], Sharon R. Lewin [3,8,9,10✉] & Joseph A. Trapani[1,2,10]

The recent dramatic appearance of variants of concern of SARS-coronavirus-2 (SARS-CoV-2) highlights the need for innovative approaches that simultaneously suppress viral replication and circumvent viral escape from host immunity and antiviral therapeutics. Here, we employ genome-wide computational prediction and single-nucleotide resolution screening to reprogram CRISPR-Cas13b against SARS-CoV-2 genomic and subgenomic RNAs. Reprogrammed Cas13b effectors targeting accessible regions of Spike and Nucleocapsid transcripts achieved >98% silencing efficiency in virus-free models. Further, optimized and multiplexed Cas13b CRISPR RNAs (crRNAs) suppress viral replication in mammalian cells infected with replication-competent SARS-CoV-2, including the recently emerging dominant variant of concern B.1.1.7. The comprehensive mutagenesis of guide-target interaction demonstrated that single-nucleotide mismatches does not impair the capacity of a potent single crRNA to simultaneously suppress ancestral and mutated SARS-CoV-2 strains in infected mammalian cells, including the Spike D614G mutant. The specificity, efficiency and rapid deployment properties of reprogrammed Cas13b described here provide a molecular blueprint for antiviral drug development to suppress and prevent a wide range of SARS-CoV-2 mutants, and is readily adaptable to other emerging pathogenic viruses.

[1] Cancer Immunology Program, Peter MacCallum Cancer Centre, Melbourne, Australia. [2] Sir Peter MacCallum Department of Oncology, The University of Melbourne, Parkville, Australia. [3] The Department of Infectious Diseases, The University of Melbourne at the Peter Doherty Institute for Infection and Immunity, Melbourne, Australia. [4] Murdoch Children's Research Institute, Parkville, Australia. [5] Children's Cancer Institute, Randwick, NSW, Australia. [6] WHO Collaborating Centre for Research and Reference in Influenza, Royal Melbourne Hospital at the Peter Doherty Institute for Infection and Immunity, Melbourne, Australia. [7] Department of Microbiology and Immunology, The University of Melbourne at the Peter Doherty Institute for Infection and Immunity, Melbourne, Australia. [8] Victorian Infectious Diseases Service, Royal Melbourne Hospital at the Peter Doherty Institute for Infection and Immunity, Melbourne, Australia. [9] Department of Infectious Diseases, Alfred Hospital and Monash University, Melbourne, Australia. [10] These authors contributed equally: Sharon R Lewin, Joseph A Trapani. ✉email: Mohamed.fareh@petermac.org; sharon.lewin@unimelb.edu.au

Severe Acute Respiratory Syndrome Coronavirus 2 (SARS-CoV-2), the cause of coronavirus disease 2019 (COVID-19) has caused >170 million infections and over 3.8 million deaths worldwide as of June 2021[1]. While a number of highly effective vaccines are now being deployed globally[2], advances in effective treatments including engineered monoclonal antibodies[3,4], and small molecule antiviral agents[5,6] have been less successful, specifically in treating severe disease or preventing progression from mild to severe disease. The rapid capacity of viruses to evolve in response to host, environmental and therapeutic pressures has been demonstrated with the global emergence of multiple new variants of concern (VOCs).

While SARS-CoV-2 possesses a moderate mutational rate due to the proof-reading activity of its RNA-dependent RNA polymerase (RdRP), we have seen the rapid emergence of new SARS-CoV-2 strains that have increased transmissibility and/or pathogenicity[7–9]. The SARS-CoV-2 D614G (Asp614Gly) variant was the first variant described. The single nucleotide substitution (A to G) in the receptor binding domain of the Spike glycoprotein, demonstrated increased affinity for the ACE2 receptor and enhanced replication capacity in vitro, potentially contributing to the global spread of this variant[8–11]. This was followed by the B.1.1.7 strain, which is now the global dominant variant of concern[12]. Analysis of virus phylogenetics globally enabled by the GISAID[13] and Nextstrain[14] databases revealed recurrent hotspot mutations in several viral subunits that likely confer selective advantage and/or alter pathogenicity[15,16]. Subsequent variants of concern including the P.1 (Brazilian) and the B.1.351 (South African) variants can evade recombinant monoclonal antibodies, antibodies in the plasma of convalescent patients, and antibodies derived from vaccinated individuals[17,18]. Some variants of concern can infect farmed animals such as minks which could eventually act as animal reservoirs and drive further mutations[19]. Therefore, there are serious concerns that the rapid evolution and spread of new SARS-CoV-2 variants of concern may compromise the global effort to control this pandemic and increase the likelihood of SARS-CoV-2 becoming endemic[20]. Taken together, there is a clear and urgent need for innovative approaches to prevention and treatment that can counteract dynamic changes in the SARS-CoV-2 genome and thwart viral escape.

The Clustered Regularly Interspaced Short Palindromic Repeats (CRISPR) Cas13 is a form of bacterial adaptive immunity that suppresses bacteriophage RNA[21–23]. Previous reports have demonstrated the potential of certain Cas13 orthologs to silence endogenous and viral RNAs in mammalian cells[24–26], however, the molecular basis by which Cas13 recognizes and suppresses replication-competent SARS-CoV-2 in infected mammalian cells, and its potential to suppress mutation-driven viral evolution and emerging variants remain to be established.

Here we report a reprogrammed CRISPR-pspCas13b that can efficiently suppress replication-competent SARS-CoV-2, including the ancestral virus, D614G and B.1.1.7 variants in monkey and human epithelial cells. The comprehensive mutagenesis analysis in this study further demonstrate that a single crRNA is resilient to single-nucleotide mismatch and remain catalytically active against mutated SARS-CoV-2 RNAs that arise in new variants. These results provide a proof-of-concept for the development of pspCas13b viral suppressors that can circumvent the evolution of SARS-CoV-2 and other pathogenic viruses.

## Results

### pspCas13b suppresses the Spike transcripts with high efficiency and specificity.
We hypothesized that the SARS-CoV-2 RNA genome and its subgenomic transcripts can be silenced with CRISPR-Cas13 in mammalian cells. We opted to use pspCas13b[27] due to its long (30-nt) spacer sequence that is anticipated to confer greater specificity than Cas13 orthologs with shorter spacer sequences.

In a proof-of concept assay, we designed Cas13 CRISPR RNAs (crRNAs) targeting predicted non-structured RNA regions of the transcript of the Spike glycoprotein, which facilitates viral invasion of host cells following binding to the ACE2 surface receptor[3,28]. A codon-optimized Spike DNA template was cloned in frame with an upstream P2A self-cleavage peptide and enhanced green fluorescent protein (eGFP), enabling co-transcription and translation of Spike and eGFP, which are separated post-translationally by P2A proteolytic self-cleavage. In this reporter assay, pspCas13b-mediated cleavage of the Spike mRNA was predicted to lead to a loss of eGFP fluorescence (Fig. 1a). We co-transfected HEK 293 T cells with the Spike-eGFP reporter plasmid together with pspCas13 linked to a blue fluorescent protein (BFP) and crRNAs targeting either the Spike transcript (four crRNAs) or non-targeting (NT) crRNA as a control. Fluorescence microscopy revealed that high silencing efficiency was achieved with all Spike-targeting crRNAs compared to the NT control (Fig. 1b, c). We achieved similar silencing in VERO cells, a kidney epithelial cell line derived from an African green monkey, commonly used in SARS-CoV-2 research due to its high susceptibility to infection[29] (Fig. 1b, c). crRNA2 achieved the highest silencing efficiency amongst the four crRNAs we tested, reaching >99% and >90% reduction in spike transcript levels in HEK 293 T and VERO cells, respectively. To demonstrate that the observed silencing was dependent on the cellular expression of crRNA, we transfected HEK 293 T cells with increasing amounts of either NT or Spike-targeting crRNA2 plasmids ranging from 0 to 104 pM (0, 0.16, 0.8, 4, and 20 ng plasmid in 100 μL volume). While NT crRNA exhibited no effect on eGFP expression, the Spike-targeting crRNA2 showed dose-dependent silencing of the Spike transcript with 50% silencing efficiency ($IC_{50}$) achieved with 5.16 pM of plasmid crRNA (equivalent to 994 pg plasmid in 100 μL of media), demonstrating that crRNA availability in the cell is key for efficient degradation of viral RNA (Fig. 1d, e). This very low $IC_{50}$ value also highlights the need for intracellular delivery of just a few copies of crRNA template plasmid to achieve effective silencing, as a result of intracellular crRNA amplification by the transcription machinery.

Like all RNA-guided nucleases, pspCas13b specificity is conferred by base-pairing between the target and crRNA spacer sequences[30–32]. Evaluating the tolerance for mismatches in this interaction is therefore critical to determining both the potential off-target activity of pspCas13b (ie, silencing cellular transcripts), and more importantly, the loss of activity against variant sequences generated by the RdRP during viral replication[33,34]. Cox et al previously examined the effect of single and double nucleotide mismatch on pspCas13b silencing using a library screen approach in a bacterial model[27], but a complementary comprehensive spacer mutagenesis study in mammalian cells using individually cloned crRNAs has not been previously investigated. We opted to determine the degree of variation in base-pairing between target RNA and the spacer of crRNA2 that could be tolerated and still lead to RNA degradation. We designed and cloned nine constructs that encoded successive 3-, 6-, or 9-nt substitutions across the whole crRNA2 spacer sequence (Fig. 1f). The nucleotide substitutions created mismatches (MSM) that perturbed the thermodynamic stability of the RNA-RNA duplex. While a 3-nt mismatch placed internally (position 14-16) or at the 3′ end (position 28-30) had minimal impact, 3-nt mismatches at the 5′ end reduced the silencing efficacy by ~50%. By comparison, 6-nt mismatches introduced at various locations markedly and consistently reduced silencing, and 9-nt mismatches completely abolished the degradation of the

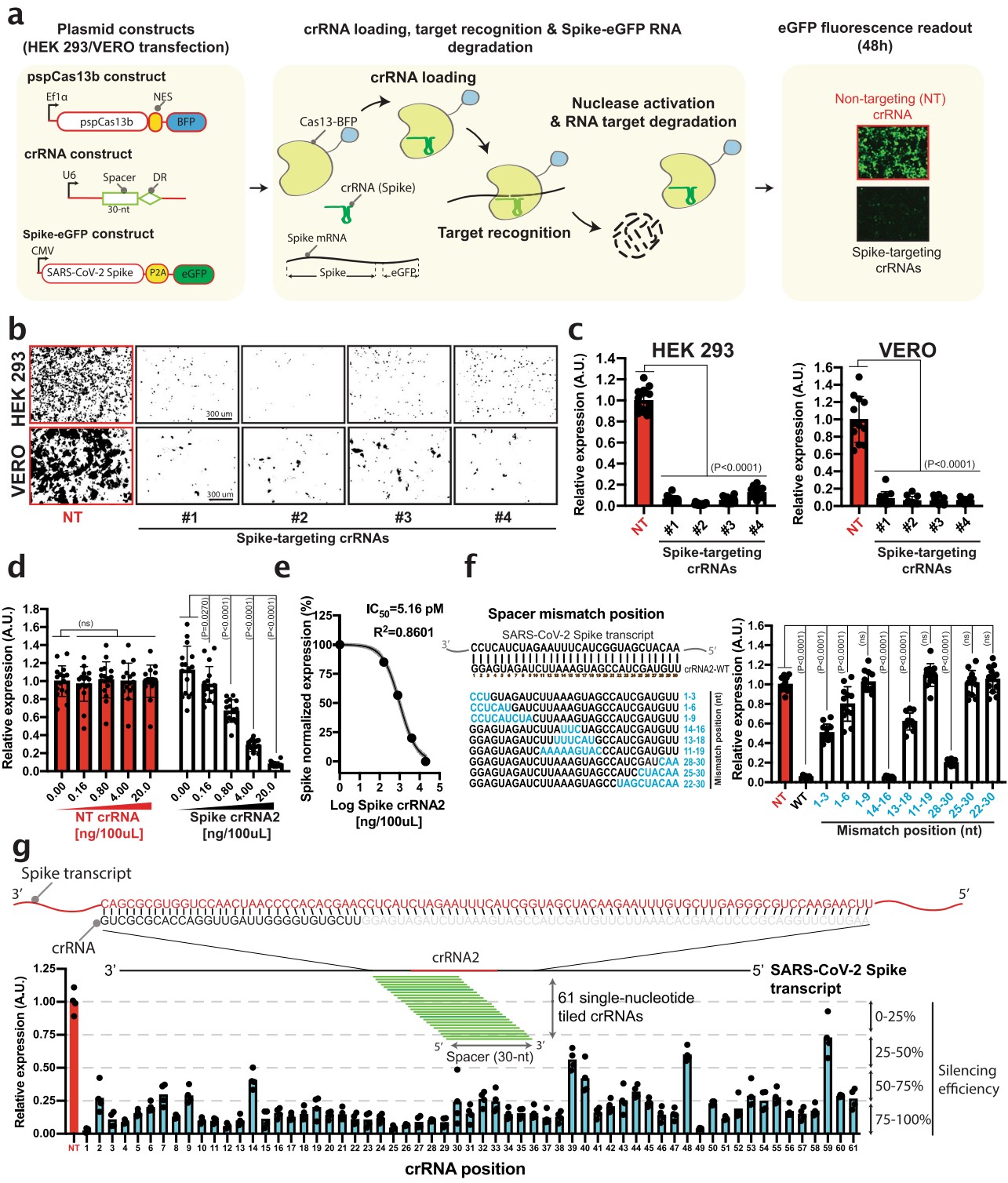

Spike transcript (Fig. 1f). Taken together, these data indicated that crRNA2 requires >21-nt base-pairing with its target to trigger the necessary CRISPR-Cas conformational change and nuclease activation necessary for target degradation[30–32], and predicts low probability of off-target activity transcriptome-wide due to the extensive (>21-nt) spacer-target base-pairing required. In contrast to the previous study that used a pooled library screen in bacteria[27], our mutagenesis in mammalian cells highlighted that up to three successive nucleotide mismatches at a central region (14–16) of the spacer are well tolerated. Importantly, the ability of pspCas13b to tolerate up to 3-nt mismatches revealed here,

especially in internal regions (14–16), indicates its potential to remain effective against the majority of variants with single-nucleotide polymorphisms in the target sequence, conferring protection against potential viral escape mutants such as the D614G mutation in the SARS-CoV-2 Spike protein[8], and mutations that compromise the efficacy of therapeutic antibodies against SARS-CoV-2[7,8,10,11,18,35,36].

Next, we asked whether the RNA target sequence and structure may influence pspCas13b silencing efficiency. To comprehensively evaluate this possibility across a defined target region, we generated 61 single-nucleotide tiled crRNAs covering the 5′ and

**Fig. 1 Reprogrammed pspCas13b silences Spike transcripts with high efficiency and specificity. a** Schematic of pspCas13b reporter assay used to track the recognition and degradation of SARS-CoV-2 Spike RNA. **b** Representative fluorescence microscopy images show the silencing of Spike transcripts with 4 targeting crRNAs in HEK 293 T (upper panel) and VERO cells (lower panel). NT is a non-targeting control crRNA; Images are processed for quantification using ImageJ. Scale bar = 300 μm. Similar results were obtained in 3 independent experiments. Unprocessed representative images are provided in the Source Data file. **c** Quantification of silencing efficiency with various crRNAs in HEK 293 T and VERO cells. Data points in the graph are normalized mean fluorescence from 4 representative fields of view per experiment imaged in $N = 3$. The data are represented in arbitrary units (A.U.). Errors are SD and $p$-values of one-way Anova test are indicated (95% confidence interval). **d** crRNA dose-dependent silencing of Spike transcripts with either NT (red) or spike targeting crRNA2 (black) in 96-well containing 100 μL of media; Data points in the graph are normalized mean fluorescence from 4 representative fields of view imaged in $N = 4$. The data are represented in arbitrary units (A.U.). Errors are SD and p-values of one-way Anova test are indicated (95% confidence interval). **e** Spike crRNA2 $IC_{50}$ value to silence 50% of Spike transcript derived from Fig. 1d. **f** Mutagenesis analysis of spacer-target interaction. The nucleotides in blue highlight residues in the spacer sequence that were mismatched with the targeted sequence. Data points in the graph are normalized mean fluorescence from 4 representative fields of view imaged in $N = 4$. The data are represented in arbitrary units (A.U.). Errors are SD and p-values of one-way Anova test are indicated (95% confidence interval). **g** Tiling of 61 crRNAs with single-nucleotide increment reveals that RNA sequence, position, and/or landscape influence pspCas13b silencing efficiency. The schematic shows the sequence of target RNA covered by 61 tiled crRNAs. Data points in the graph are normalized mean fluorescence from 4 representative fields of view imaged in N = 1 (see data from an independent biological experiment in Supplementary Figure 1). The data are represented in arbitrary units (A.U.). N is the number of independent biological experiments. Source data are provided as a Source data file.

3′ overhangs of the crRNA2 target sequence (Fig. 1g). This approach generated single-nucleotide resolution of the RNA landscape that influences pspCas13b silencing. Overall, 95% of crRNAs achieved >50% silencing efficiency, and 70% of crRNAs achieved >75% silencing of the Spike transcript. Notably, the least effective crRNAs (e.g. 39–48) and the most effective crRNAs (e.g. 15–29) were spatially clustered, suggesting that the RNA secondary structure and/or the presence of endogenous RNA binding proteins might limit spatial RNA accessibility. Similar results were obtained in an independent experiment with the same 61 tiled crRNAs (Supplementary Fig. 1). Of note, sequence analysis showed that the first 20 tiled crRNAs possess four successive G bases that are predicted to form a G-quadruplex within their spacer sequence at various locations. Despite the G-quadruplex sequence in the spacer, the majority of these crRNAs achieved very high silencing efficiency that exceeded 75%, indicating that a G-quadruplex within the spacer doesn't appear to profoundly compromise the silencing efficiency of pspCas13b. As illustrated in the probed target-spacer sequence in Fig. 1g, the tiling approach we employed, results in continuous changes in protospacer flanking sequences (PFS). For instance, single-nucleotide crRNAs between position 15 and 29 all achieved very high silencing efficiency despite the diversity in PFS sequence. Accordingly, our findings suggested that unlike other CRISPR subtypes[37], PSF or PAM-like (protospacer adjacent motif) spacers in pspCas13b didn't dominate the likelihood of silencing efficiency. Together, these data highlight the high likelihood of efficient silencing with various pspCas13b crRNAs and emphasize the flexibility of this RNA silencing tool for SARS-CoV-2 suppression.

**Silencing SARS-CoV-2 nucleocapsid transcript**. We next examined whether other SARS-CoV-2 structural and highly-conserved components essential for viral assembly can be silenced by pspCas13b. We chose to target the RNA encoding the nucleocapsid protein (NCP), a critical structural component required for viral particle assembly by packaging the RNA genome within the viral envelope[38,39]. Based on the reference genome sequence[40] we designed and cloned the original sequence of the NCP into a reporter vector in frame with a monomeric red fluorescent protein (mCherry) (Fig. 2a). We co-transfected HEK 293 T and VERO cells with the NCP-mCherry reporter plasmid together with pspCas13-BFP and a NT crRNA or three different crRNAs targeting various regions of the NCP transcript.

Based on the mCherry fluorescence intensity, we achieved high silencing efficiency in both cell lines, with all three NCP-targeting

crRNAs (Fig. 2b, c). Of these, crRNA1 exhibited the highest silencing efficiency, achieving >99% and >90% reduction in fluorescence in HEK 293 T and VERO cells, respectively. In line with the microscopy data, western blot confirmed that the three NCP-targeting crRNAs efficiently depleted NCP protein, with crRNA1 again being the most effective (Fig. 2d). As with Spike-targeting crRNAs, titration of the NCP-targeting crRNA1 plasmid (0, 0.83, 4.15, and 20.75 pM, equivalent to 0, 0.16, 0.8, 4, and 20 ng plasmid per 100 μL of media) into HEK 293 T cells demonstrated dose-dependent silencing of the target, whereas NT crRNA had no effect (Fig. 2e). The dose-dependent effect of NCP-targeting crRNA1 revealed 50% inhibition of the target ($IC_{50}$) with 0.725 pM of plasmid crRNA (140 pg in 100 μL of media in 96-well format; $R^2 = 0.906$) (Fig. 2f), whereas the $IC_{50}$ of Spike-targeting crRNA2 was 5.16 pM ($R^2 = 0.86$) (Fig. 1e). The variation in silencing efficiency (7-fold) observed with these two potent spacer sequences again highlighted that affinity and target accessibility need to be considered when selecting the optimal crRNA for maximal target silencing.

Next, we examined the extent to which mismatches in the spacer-target RNA-RNA hybrid of the highly efficient NCP-targeting crRNA1 would compromise its silencing potency. We designed 29 additional crRNA constructs that incorporated mismatches at the 5′ end, 3′ end, or at internal positions (blue residues, Fig. 2g–i). Mismatch length varied from 3 to 30 nucleotides to cover the entire spacer sequence. At the 5′ end, mismatches at positions 1–3 or 1–6 were well-tolerated and caused minor reductions in silencing efficiency in HEK 293 T cells transfected with 20 ng of crRNAs, while altering positions 1-9 reduced silencing by ~30%. However, by titrating the quantity of crRNA delivered, we revealed a noticeable loss of silencing efficiency when 1–3, 1–6, or 1–9 mismatches were introduced (Supplementary Fig. 2). Mismatches longer than 9-nt at the 5′ end completely abrogated silencing. Similarly, 3-nt mismatches placed internally (positions 14–16) or at the 3′ end were well tolerated, while 6-nt mismatches reduced silencing by ~30%. Introducing >6-nt mismatches at both internal and 3′ end positions caused a complete loss of silencing (Fig. 2h, i). Together, these mutagenesis data again highlighted that pspCas13b can tolerate mismatches, and the variability in tolerance between various spacer sequences may be proportional to spacer-target affinity.

**pspCas13b silencing tolerates a single-nucleotide mismatch with the target**. Viral evolution takes place predominantly through single-nucleotide indels brought about by error-prone polymerases. We investigated whether pspCas13b can tolerate a

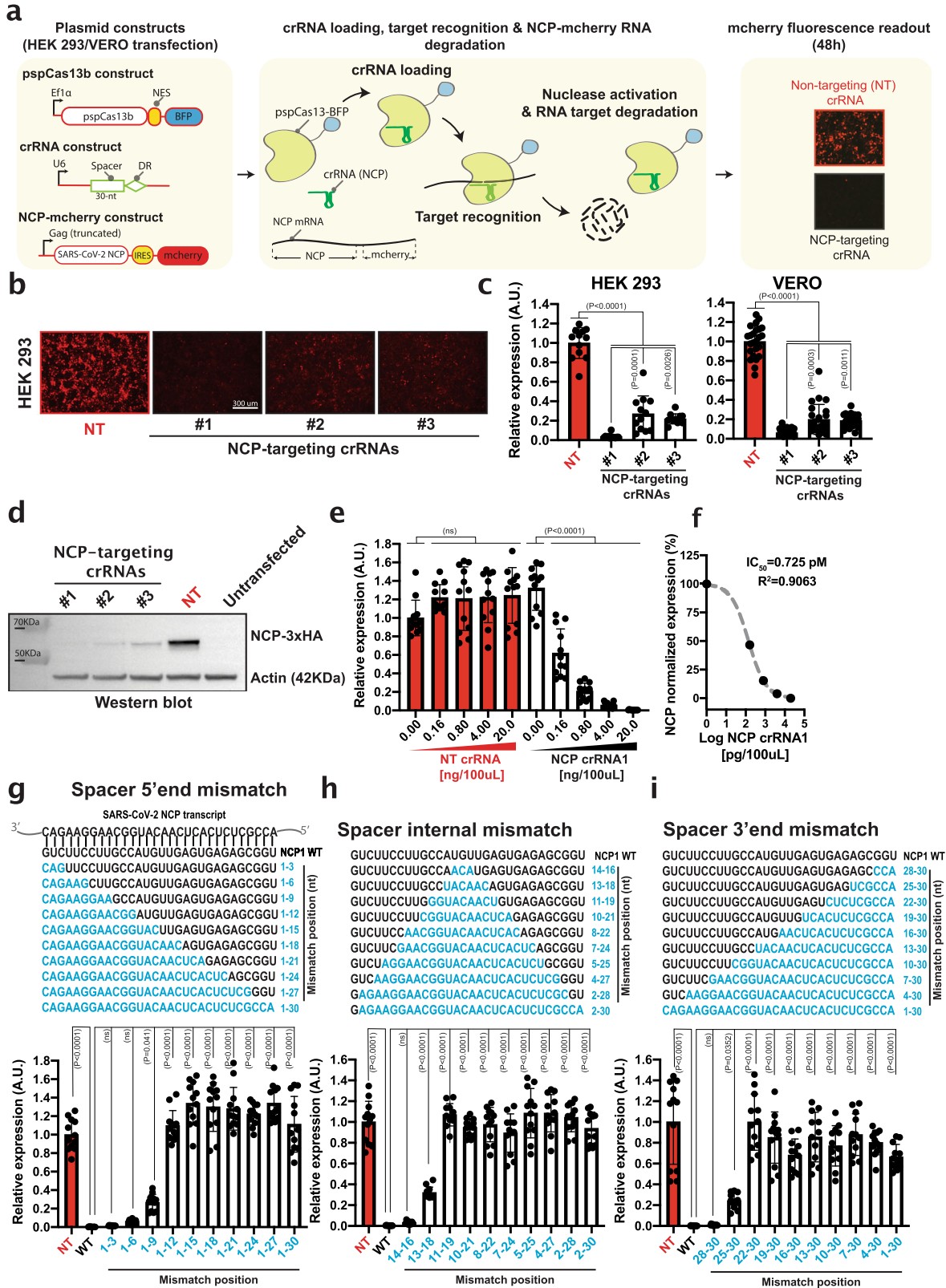

single-nucleotide mismatch between the spacer and the SARS-CoV-2 RNA target, potentially enabling our silencing technology to remain effective against spontaneous mutations. We re-designed the crRNAs targeting either the Spike (crRNA2) or the nucleocapsid (crRNA1) transcripts to harbour a single-nucleotide mismatch at spacer positions 1, 5, 10, 15, 20, 25, and 30 and compared their silencing efficiency to their fully-matched

wildtype counterparts. Overall, we found that a single-nucleotide mismatch at various locations of the spacer was well tolerated, with no appreciable impact on target silencing (Fig. 3a, b). Among all the constructs we tested, only a single nucleotide mismatch in the first nucleotide of the spacer sequence of crRNA1 targeting Nucleocapsid exhibited a moderate loss of silencing (Fig. 3b). In line with the fluorescence data, western blot

**Fig. 2 pspCas13b suppresses SARS-CoV-2 nucleoprotein (NCP) transcripts with high efficiency and specificity. a** Schematic of pspCas13b reporter assay to monitor NCP silencing with various pspCas13b crRNAs. **b** Representative fluorescence microscopy images show the silencing efficiency of the NCP transcripts with 3 targeting crRNAs in HEK 293 T. NT is a non-targeting control crRNA. Scale bar = 300 μm. Similar results were obtained in 3 and 6 independent experiments in HEK 293 T and VERO cells, respectively. Unprocessed representative images are provided in the Source Data file. **c** The histograms quantify the silencing efficiency of various crRNAs in HEK 293 T ($N = 3$) and VERO cells ($N = 6$). Data points are normalized mean fluorescence from 4 representative fields of view per condition imaged. The data are represented in arbitrary units (A.U.). Errors are SD and p-values of one-way Anova test are indicated (95% confidence interval). **d** Western blot analysis of NCP silencing efficiency in HEK 293 T obtained with various crRNAs ($N = 3$; uncropped blots from 3 independent biological experiments are provided in Source data). **e** crRNA dose-dependent silencing of the NCP transcripts with either NT (red) or NCP targeting crRNA1 (black) in 96-well containing 100 μL of media. Data points are normalized mean fluorescence from 4 representative fields of view per condition imaged; $N = 3$. The data are represented in arbitrary units (A.U.). Errors are SD and p-values of one-way Anova test are indicated (95% confidence interval). **f** NCP crRNA1 IC50 value (pM) to silence 50% of the Nucleocapsid transcript derived from Fig. 2E. **g-i** Comprehensive analysis of spacer-target interaction, examining specificity and mismatch tolerance of pspCas13b crRNA at various positions of the spacer. The nucleotides in blue highlight mismatch positions in the spacer sequence. Data points in the graph are mean fluorescence from 4 representative field of views per condition imaged; $N = 3$. The data are represented in arbitrary units (A.U.). Errors are SD and p-values of one-way Anova test are indicated (95% confidence interval). $N$ is the number of independent biological replicates. Source data are provided as a Source data file.

analysis of Nucleocapsid protein expression confirmed that single-nucleotide mismatch at position 5, 10, 15, 20, 25, and 30 was well tolerated and full silencing efficiency was retained, while a mismatch at position 1 led to a partial loss of silencing (Fig. 3c). These data highlight the potential of a single pspCas13b crRNA to silence SARS-CoV-2 variants and overcome mutation-driven viral escape.

**pspCas13b suppresses SARS-CoV-2 virus replication in infected cells.** A recent study attempted to target a SARS-CoV-2 RNA construct with a Cas13d ortholog but the silencing of live replication-competent SARS-CoV-2 was not examined[26]. Rather, a genetically engineered H1N1 influenza strain expressing a fluorescent reporter was used. To determine whether our reprogrammed pspCas13b could inhibit SARS-CoV-2 RNA replication, we transfected VERO cells with pspCas13b-BFP and either NT crRNA or the crRNA1 that most efficiently reduced Nucleocapsid RNA levels in virus-free models (Fig. 2). We optimised transfection conditions of VERO cells to achieve 25–40% apparent transfection efficiency based on BFP expression and detection threshold in VERO cells as measured by flow cytometry analysis (Supplementary Fig. 3). After 48 h post-transfection, we infected the cells with replication-competent SARS-CoV-2 at a Multiplicity Of Infection (MOI) of 0.01 or 0.1, and assayed the culture medium at 1 h, 24 h, and 48 h time-points using real-time PCR (RT-PCR) for viral RNA[29]. We also determined the titer of infectious SARS-CoV-2 by exposing fresh VERO cells to the supernatant and quantifying the cytopathogenic effect (CPE) by light microscopy and manual counting to determine the tissue culture infectious dose that inhibits 50% of virus growth (TCID50) (Fig. 4a). In cells transfected with NT crRNA and infected at 0.1 MOI, viral RNA in the supernatant increased ~15-fold at 24 h compared to baseline and this level was maintained at 48 h. By contrast, in the presence of NCP-targeting crRNA1 there was a marked reduction in viral RNA detected. RT-PCR analysis showed that crRNA1 reduced the viral load by ~90% and 60% at 24 h and 48 h post-infection, respectively, as compared to the NT crRNA (Fig. 4b). However, NCP-targeting crRNA1 did not completely abrogate viral replication in this biological system, as the viral titer still increased 3-fold between 24 h and 48 h (Fig. 4b). This partial inhibition of SARS-CoV-2 replication by pspCas13b is likely explained by a combination of a high viral load (0.1 MOI) that may have saturated intracellular pspCas13b nucleoproteins, and the limited apparent transfection efficiency in VERO cells (25-40%) (Supplementary Fig. 3). Subsequently, viral replication may take place in untransfected cells or cells expressing low levels of pspCas13b and crRNA. Increasing the efficiency

of delivery of pspCas13b and its crRNA should result in even greater viral suppression.

We therefore infected VERO cells with a lower viral titer (0.01 MOI). Viral RNA increased in the supernatant of cells expressing NT crRNA by ~14-fold 48 h post-infection. By contrast, viral RNAs in the culture medium of cells expressing NCP-1 crRNA remained at the basal level (Fig. 4b), demonstrating that pspCas13b and NCP-targeting crRNA1 efficiently suppressed viral replication in infected mammalian cells. Consistent with RT-PCR, the infectivity assay confirmed that the NCP-targeting crRNA1 greatly reduced the release of infectious virus (measured as TCID50/mL) in the supernatant (Fig. 4c). The high suppression of viral replication with various crRNAs despite the apparent moderate transfection efficiency (25–40%) indicates that the transfection efficiency was possibly underestimated. We postulate that some cells may express low levels of Cas13-BFP that was below the detection threshold, or the BFP may exhibit a shorter half-life compared to Cas13 and therefore contributed to the underestimation of transfection efficiency. Overall, these data indicate that pspCas13b can achieve efficient SARS-Co-2 suppression despite the apparent partial cell delivery and expression.

The emergence of SARS-CoV-2, potentially with enhanced transmissibility and pathogenicity, poses a serious challenge to any single antiviral or antibody therapy[7,17]. Likewise, targeting SARS-CoV-2 with a single crRNA may have limited efficacy due to viral adaptation and/or poor accessibility to viral RNA in vivo due to intrinsic RNA folding and/or interference from RNA-binding proteins. We postulated that simultaneously targeting several regions of SARS-CoV-2 viral RNAs with multiplexed crRNAs would mitigate these effects, and reduce the likelihood of viral adaptation. We transfected VERO cells with four pools of crRNAs each containing four different crRNAs targeting either SARS-CoV-2 structural proteins Spike and NCP, or the non-structural proteins NSP7 and NSP8 that act as co-factors critical for the RdRP[6,33,34]. We again infected the VERO cells with SARS-CoV-2, and quantified viral RNA and infectious virus in the culture supernatant. We found that all four crRNAs pools markedly reduced viral RNA (Fig. 4d), and infectious virus (Fig. 4e) in the supernatant at both 0.1 and 0.01 MOI.

Likewise, when we challenged VERO cells with 0.1 and 0.01 MOI of SARS-CoV-2 virus 72 h post-transfection of pspCas13b and various crRNAs, we achieved ~80% and ~90% suppression of viral RNAs in the supernatant with all tested crRNAs, respectively (Fig. 4f & Supplementary Fig. 4). These results indicate that SARS-CoV-2 viral suppression from a single pspCas13b transfection can persist beyond 5 days post-transfection, and 3 days post-infection. Of note, deploying pooled crRNAs is a conservative

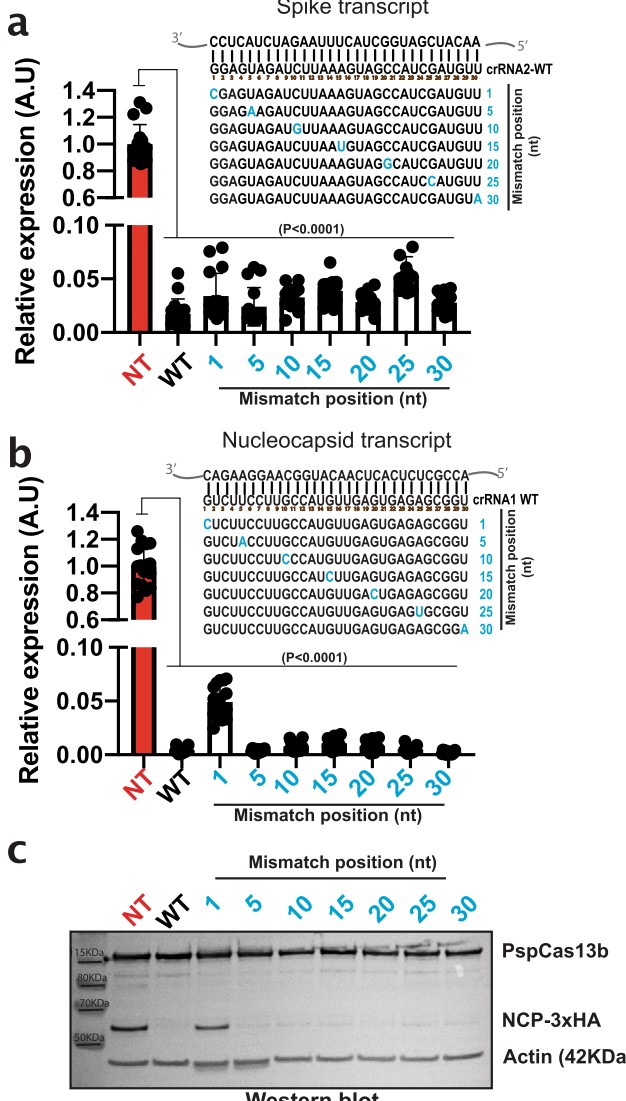

**Fig. 3 pspCas13b silencing tolerates single-nucleotide mismatch with SARS-CoV-2 targets.** Fluorescence-based reporter assays to assess the silencing efficiency of Spike (**a**) or Nucleocapsid (**b**) transcripts with crRNAs harbouring single-nucleotide mismatch at various location of the spacer-target interface 48 h post-transfection. Nucleotides in blue highlight the mismatch position. Data points in the graphs are mean fluorescence from 4 representative fields of view per condition imaged; $N = 4$. The data are represented in arbitrary units (A.U.). Errors are SD and p-values of one-way Anova test are indicated (95% confidence interval). **c** Representative Western blot analysis to examine the expression level of Nucleocapsid protein (3xHA tagged) in HEK 293 T cells expressing NCP-1 crRNA with single-nucleotide mismatch at various positions of the spacer-target interface 24 h post-transfection; $N = 3$. (See uncropped blots from 3 independent biological experiments in Source file). N is the number of independent biological replicates. Source data are provided as a Source data file.

approach to mitigate the risk of viral escape in the event of mutations occurring within the targeted sequence of a single crRNA (e.g. deletion or substitution of several bases), or in the event of target inaccessibility due to SARS-CoV-2 RNA folding and interactions with RNA binding proteins.

In addition to VERO cells (monkey's origin), we questioned whether we can demonstrate SARS-CoV-2 suppression with pspCas13b in a human epithelial cell line. Calu-3 cells are derived

from lung adenocarcinoma and are commonly used in SARS-CoV-2 research due to their SARS-CoV-2 susceptibility[41]. We transfected Calu-3 cells with pspCas13b and either NT or NCP-1 crRNA 48 h prior to SARS-CoV-2 infection (MOI 0.1), and monitored the silencing efficiency over time by TCID50 infectivity assay at time points 1, 24, and 48 h. The NT crRNA data indicated slow kinetics of infection and/or viral replication in Calu-3 compared to VERO cells, as the viral titer initially dropped at 24 h before increasing by 10-fold between 24 and 48 h. However, SARs-CoV-2 replication remained very low (basal level) 48 h post-infection in Calu-3 cells expressing NCP-1 crRNA (Fig. 4g). This data indicates that pspCas13b can silence SARS-CoV-2 in human epithelial cells that mimics natural host cells of SARS-CoV-2.

We also sought to show whether Cas13 can silence other variants of SARS-CoV-2. We transfected VERO cells with Cas13 and either NT or NCP-1 crRNA before infecting them with 0.01 MOI of the highly infectious and virulent B.1.1.7 variant (known as the UK strain). The NT crRNA group showed a drastic increase in viral replication at 72 h post-infection reaching a 13- and 17-fold increase between 48- and 72-hours in infectivity (TCID50) and RT-PCR assays, respectively. Conversely, NCP-1 crRNA largely limited B.1.1.7 variant replication in VERO cells in RT-PCR and infectivity (TCID50) assays (Fig. 4h). Thus, demonstrating that a single pspCas13b crRNA targeting a conserved region of nucleoprotein can silence SARS-CoV-2 variants.

To the best of our knowledge, together with a recently published report by Blanchard et al.[42], these are the first two studies that definitively demonstrate the effective and specific targeting of replication-competent SARS-CoV-2 in infected mammalian cells using different CRISPR-Cas13 orthologs, and the first demonstration of SARS-CoV-2 variants suppression with the CRISPR-pspCas13b ortholog.

**pspCas13b remains resilient against single-nucleotide mismatches in the RNA of SARS-CoV-2 variants.** A single nucleotide substitution in the receptor binding domain (RBD) of the Spike protein led to the global emergence of the SARS-CoV-2 D614G variant with increased ACE-2 affinity and infective potential[8,10]. Based on the single-nucleotide mismatch data described above (Fig. 3a, b), we hypothesized that crRNAs designed against ancestral Spike D614 genomic sequence should remain effective against the D614G mutant RNAs despite a single-nucleotide mismatch at the spacer-target interface. To test this, we cloned part of the D614G Spike coding sequence (1221 bp; D614G SARS-CoV-2 genomic region between 22,760—23,980) that hasn't been codon-optimized into the mCherry reporter system, and assessed the silencing efficiency of six tiled crRNAs targeting this frequently mutated hotspot. The spacer sequences of all six tiled crRNAs tested were designed to fully match the ancestral Spike sequence (D614) and harbour one nucleotide mismatch (G–U mismatch) with the Spike transcript of D614G mutant at spacer positions 5, 10, 15, 20, 25, and 30 (Fig. 5a). We found that all six crRNAs significantly degraded the D614G mutant transcript (P < 0.0001). Interestingly, crRNAs with a single-nucleotide mismatch at spacer position 15, 20, 25, or 30 were highly effective and exhibited >95% silencing efficiency against the D614G mutant transcript. By contrast, crRNAs with a mismatch at position 5 or 10 showed reduced silencing efficiency estimated at approximatively 65—88% removal of the D614G mutant transcript (Fig. 5a). G–U mismatch created by these crRNAs targeting the D614G region can still base-pair through a G–U RNA wobble[43], which may stabilize the RNA-RNA duplex and mask the impact of single-nucleotide mismatch on

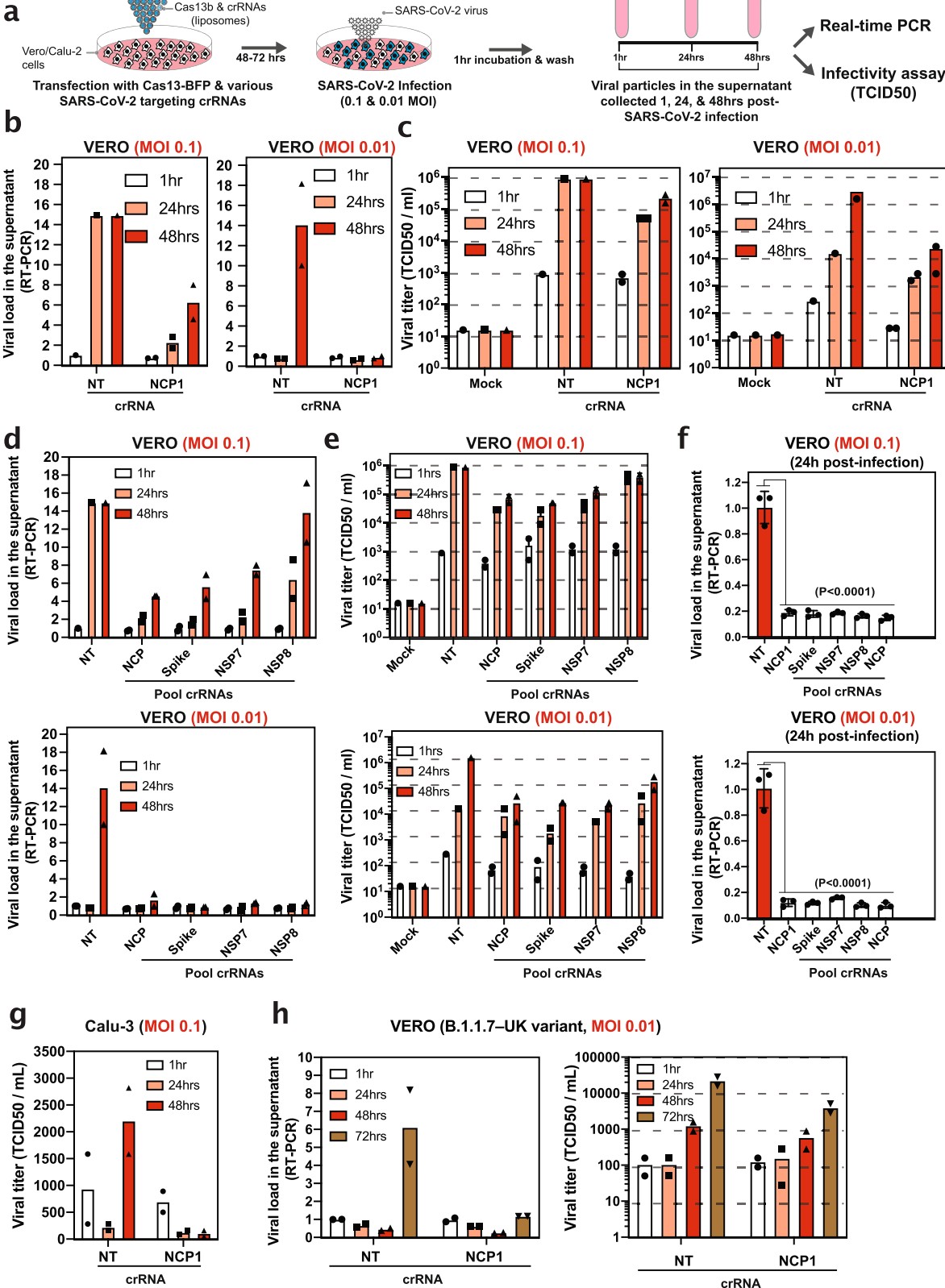

pspCas13b silencing. To test this possibility, we tested an additional six crRNAs harbouring a G–G mismatch with the targeted D614G sequence, and six crRNAs with a full match with the target through G–C match that served as a control (Fig. 5b, c). Overall, U or G at various mismatch positions did not dramatically change the silencing efficiency compared to the reference fully matched crRNAs (with a C), confirming that a single-base

mismatch per spacer was well tolerated by pspCas13b. We observed a moderate reduction in silencing efficiency when a G was introduced at position 15 and 30 that led to the removal of 74% and 81% of D614G transcript, respectively. Conversely, a G mismatch at position 5 improved the silencing efficiency compared to U or C at the same position (Fig. 5a–c). A single-nucleotide polymorphism between these spacers slightly

**Fig. 4 Silencing of SARS-CoV-2 virus replication in VERO cells. a** Schematic of infection assay to assess pspCas13b-mediated suppression of SARS-CoV-2 replication in VERO cells. VERO cells were transfected with liposomes containing pspCas13b-BFP and various crRNA constructs. 48-72 h post-transfection, cells were infected with SARS-CoV-2 for 1 h and the kinetics of viral replication were assessed in supernatants collected at 1, 24, and 48 h post-infection for production of viral RNA using RT-PCR and infectious virus by quantification of the TCID50 on VERO cells. **b** RT-PCR and **c** infectivity assays to evaluate the kinetic of viral replication in VERO cells expressing either NT or NCP-targeting crRNA1 48 h post-transfection of pspCas13b and crRNAs transfection, N = 2. **d** RT-PCR and **e** infectivity assays to monitor the kinetics of viral replication in VERO cells expressing either NT or various pools of crRNA targeting NCP, Spike, NSP7, and NSP8. VERO cells were infected with a replication-competent SARS-CoV-2 48 h post-transfection of pspCas13b and crRNAs; N = 2. **f** RT-PCR analysis of the viral load in the supernatant of VERO cells expressing either NT, NCP-1, or various pools of crRNA targeting NCP, Spike, NSP7, and NSP8. VERO cells were infected with a replication-competent SARS-CoV-2 48 h post-transfection of pspCas13b and crRNAs; N = 3; Data are normalized means and errors are SD; Results analysed with one-way Anova test; (95% confidence interval). **g** Infectivity assay to monitor the kinetics of viral replication in Calu-3 cells expressing either NT or NCP-1 crRNA targeting the nucleoprotein. Calu-3 cells were infected with a replication-competent SARS-CoV-2 (MOI 0.1) 48 h post-transfection of pspCas13b and crRNAs; N = 2. (**h**) Infectivity assay to monitor the kinetics of B.1.1.7 SARS-CoV-2 replication (variant first identified in the UK) in VERO cells expressing either NT or NCP-1 crRNA targeting the nucleoprotein (NCP). VERO cells were infected with a replication-competent B.1.1.7 variant (MOI 0.1) 48 h post-transfection of pspCas13b and crRNAs and viral loads monitored at 1, 24, 48, and 72-h post-infection through RT-PCR (left) and infectivity assay (right); N = 2. N is the number of independent biological experiments. All source data in this figure are provided as a Source data file.

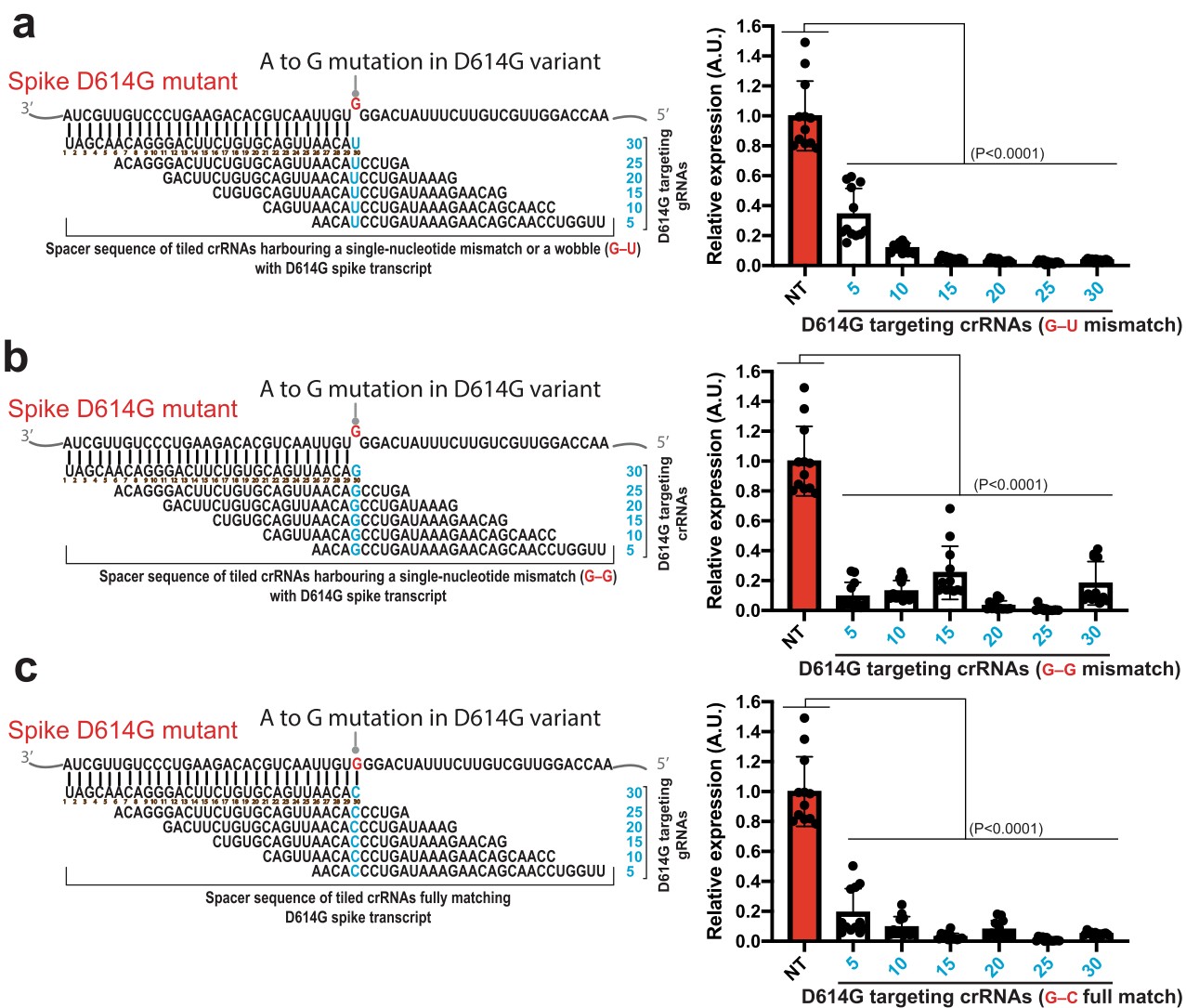

**Fig. 5 Single crRNAs targeting the D614 genomic mutation hotspot can silence D614G variant's RNA through the single-nucleotide mismatch tolerance. a–c** Fluorescence-based reporter assay to assess the silencing efficiency of the D614G Spike transcripts with 18 tiled crRNAs harbouring a G–U (**a**), G–G (**b**) single-nucleotide mismatch, or G–C full match (**c**) with the target at various spacer positions; Data points in the graphs are mean fluorescence from 4 representative fields of view per condition imaged; N = 3. The data are represented in arbitrary units (A.U.). Errors are SD and p-values of one-way Anova test are indicated (95% confidence interval). N is the number of independent biological experiments. Source data are provided as a Source data file.

modulated their silencing efficiencies possibly through subtle changes in crRNA transcription, folding, and pspCas13 affinity. Overall, these data indicate that a single-nucleotide mismatch at various positions is well-tolerated and doesn't alter the silencing efficiency of a given crRNA. Together, these data suggest that pspCas13 and the D614G targeting crRNAs with a single-nucleotide mismatch are likely to remain effective against both ancestral and D614G mutant in infected cells. Thus, the mismatch tolerance molecular mechanism described here is expected to confer pspCas13b resilience against spontaneous single-nucleotide mutations that drive viral escape in SARS-CoV-2 and other pathogenic viruses.

**pspCas13b suppresses mutation-driven SARS-CoV-2 evolution**. In the virus-free reporter model, we demonstrated that pspCas13b crRNAs targeting the D614 genomic region remained effective against the spike transcript of D614G variant despite a G–U mismatch at various positions (Fig. 6a). To test the mismatch tolerance of pspCas13b system against various replication-competent SARS-CoV-2 viral strains, we transfected VERO cells with NT crRNA, NCP-1 targeting crRNA (positive control), or 4 tiled crRNAs (10, 15, 20, 30) fully matching the ancestral Spike sequence D614 but therefore containing a single-nucleotide mismatch (G–U) with the D614G variant (see the schematic illustration in Fig. 6a, b). 48 h post-infection, VERO cells expressing pspCas13-BFP and various crRNAs were infected with either the ancestral or D614G mutant SARS-CoV-2, and the viral loads in supernatants were quantified by RT-PCR 1 h (to determine the initial viral input of both strains), 24 h and 48 h post-infection (Fig. 6b). The comparison of viral loads in the supernatant of cells infected with either the ancestral or D614G in the control groups (NT) showed 7.7, 4.2, and 2.3-fold higher viral loads in the D614G samples at timepoints 1 h, 24 h, and 48 h, respectively (Fig. 6c, e and g). These data showed that the initial D614G viral load used here was more than 7 times higher than the ancestral strain.

As anticipated, all SARS-CoV-2 targeting crRNAs tested showed no significant viral suppression against the ancestral nor the D614G strains 1-hour post-infection. The absence of viral suppression with targeting crRNAs at this early timepoint of infection was anticipated because pspCas13b-mediated viral suppression requires intracellular viral replication that had not yet occurred at 1-hour timepoint post-infection.

In the positive control, NCP-1 crRNA was used to target a fully matching sequence within the Nucleocapsid RNA that is conserved in both the ancestral and D614G strains. Consistent with the data in Figs. 2 and 4, the NCP-1 crRNA again showed a very high silencing efficiency, and suppressed 84−88% and 73−74% of replication-competent ancestral and D614G viruses, respectively, at both the 24 and 48 h timepoints (Fig. 6f, h). The moderate loss of viral suppression in cells infected with the D614G mutant was likely attributable to the 7-fold higher initial viral titer (Fig. 6c, e & g). Among the four tiled crRNAs targeting the ancestral D614 position in the Spike RNA, three of them (15, 20, and 30) showed efficient viral suppression reaching 78−84% and 44−60% in cells infected with the ancestral and D614G strains respectively, at both 24 and 48 h timepoints (Fig. 6f & h). Again, the moderate reduction in suppression of the D614G strain with these three crRNAs was likely attributed to the initial higher viral titer of this D614G strain. Last, the fourth crRNA we tested (harbouring a mismatch with the D614G mutation at spacer position 10) showed the lowest affinity and viral suppression potential with 62−71% suppression of the ancestral strain, and only 3−7% suppression of the D614G, which was not statistically significant at 24 nor at 48 h (Fig. 6f, h). This result is consistent with the data obtained from the virus-free model (Fig. 6a), where the crRNA harbouring a single-nucleotide mismatch with D614G at spacer position 10 exhibited the lowest silencing efficiency among the 4 crRNAs tested. This correlation highlights the predictive power of the virus-free model we developed, and emphasises its utility to tractably screen and select the best performing crRNAs. Overall, the data in Fig. 6 provide strong evidence that optimized single crRNAs are likely to retain efficacy against spontaneous point-mutations that arise during viral replication and can efficiently silence ancestral and emerging SARS-CoV-2 strains, including the D614G.

**In silico prediction and design of a genome-wide SARS-CoV-2 crRNA library**. Based on the high silencing efficiency of SARS-CoV-2 in virus-free and infection models shown above, we propose that a genome-wide SARS-CoV-2 crRNA library for pspCas13b would be a useful tool to silence various SARS-CoV-2 subunits and interrogate their role in the biology of this new virus.

We used the first publicly available genomic sequence of SARS-CoV-2 as the reference genome[40] to design pspCas13 crRNAs, and developed a bioinformatics workflow for in silico design of 29,894 single-nucleotide tiled crRNAs covering the entire genome, from 5′ to 3′ end (Supplementary Fig. 5 & Supplementary Data file 1). We further refined the list of crRNAs by excluding (i) spacer sequences with poly-T or poly-C repeats (four or more successive T or C nucleotides) that are anticipated to prematurely terminate the Pol III-driven transcription of crRNAs or target G-quadruplex structure, respectively (Supplementary Data file 2), (ii) crRNAs harbouring four successive C that target RNA sequences with G-quadruplex, and (iii) spacers or target sequences with predicted thermodynamically stable RNA-RNA duplexes that may hinder crRNA loading into pspCas13b and/or compromise target accessibility (Supplementary Data file 3).

To experimentally probe whether crRNAs with poly-T and poly-C repeats (>3 T or >3 C) have lower silencing efficiency, we designed additional crRNAs targeting the NCP-reporter transcript that possessed successive 4 T, 5 T, or 4 C repeats at various positions of the spacer. (Supplementary Fig. 6a–c). All crRNA harbouring 5 T yielded poor silencing efficiency that was lower than ~50%, and six out of eight crRNAs with successive 4 T we tested yielded very poor silencing efficiency compared to NCP-1 crRNA that lacked successive poly-T sequences (Supplementary Fig. 6a, b). Overall, poly-T sequences within the spacer decreased the likelihood of efficient silencing with pspCas13b, likely due to prematurely terminated transcription[44].

Similarly, crRNAs with 4 C that targeted predicted structured G-quadruplex sequences also yielded limited silencing efficiencies. Six out of nine crRNAs harbouring a 4 C in the spacer we tested (66.6%) showed poor silencing efficiency, while the other three crRNAs (33.3%) exhibited moderate to good silencing efficiency. These data indicated that G-quadruplex sequences on the target may also reduce the likelihood of efficient silencing with pspCas13b, possibly due to limited target accessibility (Supplementary Fig. 6c). Together, these data indicate that the exclusion of crRNA with poly-T (> 3 T) and poly-C (> 3 C) sequences in the spacer from the SARS-CoV-2 crRNAs libraries is likely to improve overall silencing efficiency.

After filtering the SARS-CoV-2 crRNAs library by excluding spacers with the aforementioned undesirable characteristics, we identified 838 spacer sequences that best satisfied the selection criteria (Supplementary Data file 3). As all 838 30-nt spacer sequences fully base-pair with the reference SARS-CoV-2 RNA, all are predicted to achieve good targeting efficiency. Given

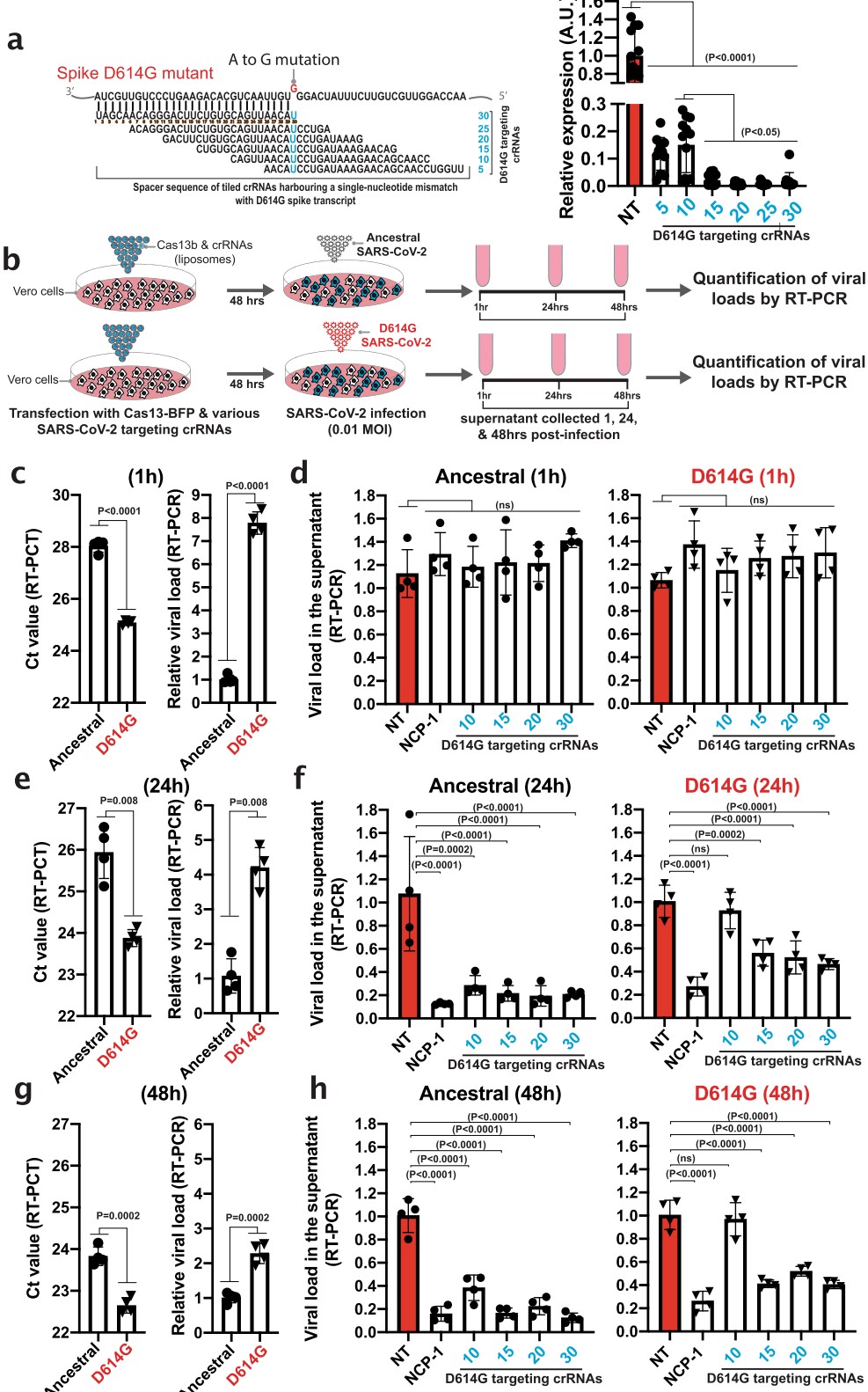

pspCas13b targeting appears to be dependent on multiple parameters, some of which remain unknown, we cannot definitively exclude the possibility that some crRNAs in the library may still yield suboptimal silencing. Therefore, we recommend experimental validation the pre-selected crRNAs in virus-free or infection models.

We subsequently used publicly available genomic data from GISIAD[13] and the UCSC genome browser to map all the 838 crRNAs to their target sites on SARS-CoV-2 genome. We identified crRNAs targeting either transcribed subgenomic or intergenic regions (Supplementary Fig. 7). The predicted top crRNAs list contained 827 crRNAs targeting subgenomic RNAs

**Fig. 6 Single crRNAs targeting the D614 genomic mutation hotspot can silence both ancestral and D614G variants in infected cells through single-nucleotide mismatch tolerance. a** Fluorescence-based reporter assay to assess the silencing efficiency of the D614G Spike transcripts with 6 tiled crRNAs harbouring a single-nucleotide mismatch with the target at various spacer positions; Data points in the graphs are mean fluorescence from 4 representative fields of view per condition imaged; $N = 3$. The data are represented in arbitrary units (A.U.). Errors are SD and p-values of one-way Anova test are indicated (95% confidence interval). **b** Schematic of infection assays to assess pspCas13b-mediated suppression of both ancestral and D614G replication-competent SARS-CoV-2 in infected VERO cells. VERO cells were transfected with pspCas13b-BFP and various crRNA constructs. 48 h post-transfection, cells were infected with either ancestral or D614G SARS-CoV-2 for 1 h, and the kinetics of viral replication was assessed in supernatant collected at 1 h (initial viral load), 24 h, and 48 h post-infection via quantification of viral RNA in the supernatant using RT-PCR. **c, e, g** RT-PCR assays to evaluate relative viral loads (ancestral versus D614G) in control VERO cells expressing non-targeting (NT) crRNA at timepoints 1 hr to estimate the initial viral input (**c**), 24 h (**e**), and 48 h (**g**) post-infection, $N = 4$; Data are normalized means and errors are SD; Results are analysed by unpaired two-tailed Student's t-test (95% confidence interval). (**d, f, h**) RT-PCR assays to evaluate the suppression of ancestral and D614G SARS-CoV-2 strains in VERO cells expressing either non-targeting (NT), NCP-1, or tiled crRNAs targeting the D614G mutation hotspot in the Spike genomic and subgenomic RNA 1 h (**d**), 24 h (**f**), and 48 h (**h**) post-transfection, $N = 4$; Data are normalized means and errors are SD; Results are analysed by one-way ANOVA test (95% confidence interval). N is the number of independent biological experiments. Source data are provided as a Source data file.

and 11 crRNAs targeting intergenic regions. We speculate that targeting subgenomic regions that are transcribed may achieve higher viral suppression due to target transcript abundance.

Mutation hotspots refer to genomic regions that are frequently mutated in SARS-CoV-2 genomes[16]. We hypothesized that avoiding these mutation hotspots would reduce the probability of viral escape from pspCas13b silencing in the unlikely event of deletion, insertion, or substitution of several bases within these regions. Therefore, we highlighted in the top hit library (red) crRNAs that target these frequently mutated genomic hotspots. Overall, there were 18 mutation hotspots (>0.04 mutation frequency) genome wide, and only 21 crRNAs in the top hit list overlapped with these mutation hotspots. We recommend avoiding using these crRNAs for therapeutic purposes due to the risk of multiple mismatch occurrence within these hotspot regions that may impair the silencing efficiency.

Some crRNAs in the top hit list may cause off-target silencing of cellular transcripts if there is extensive sequence complementarity with transcripts in the human transcriptome. We sought to bioinformatically predict this potential off-target activity of cellular transcripts by aligning the crRNAs in the top hit list to the human transcriptome. We used Burrows-Wheeler Alignment tool (BWA)[45] and allowed for various numbers of mismatches between the space and the target. Interestingly, we found no crRNA with complete match, 1, and 2 nucleotide mismatches within the entire human transcriptome. Potential off-target activity with 3, 4, 5, and 6 nucleotide mismatches were found in 6, 21, 61, and 344 crRNAs, respectively (Supplementary Data file 4 and 5). We took a very conservative approach and excluded any crRNA that had six nucleotide or fewer mismatches with the human transcriptome. This stringent filtering strategy identified 344 crRNAs in the top hits with potential off-target activity, and reduced the number of crRNAs from 838 to 494 (Supplementary Data file 4 and 5). Importantly, the unique 30-nt spacer length of the pspCas13b ortholog enables such stringent filtering to minimize the impact of potential off-targeting risk, whereas other Cas13 orthologs with shorter spacer sequences may have greater off-target activity as specificity is conferred by spacer-target base-pairing and proportional to the length of the spacer. We acknowledge that this bioinformatic prediction of off-target activity requires further validation experimentally using transcriptome analyses with various crRNAs in viral infection and silencing settings.

This in silico predictive workflow and database represents a valuable investigative tool for the scientific community to interrogate SARS-CoV-2 biology through efficient silencing of various SARS-CoV-2 subunits, and may aid in the development of pspCas13b-based anti-SARS-CoV-2 therapeutics.

## Discussion

The remarkable capability of RNA viruses to adapt to selective host and environmental pressure is highly dependent on their ability to generate genomic diversity through the occurrence of de novo mutations[46]. Mutation-driven viral evolution can generate drug resistance, immune escape, and increased efficiency of transmission and pathogenicity, all of which are detrimental to the host. Although our understanding of SARS-CoV-2 mutation-driven escape mechanisms remains limited, the emergence of new variants, which possess increased infective potential[8] or are resistant to recombinant monoclonal antibodies and antibodies in the sera of convalescent patients and vaccinated individuals[7,8,17,18,36] are of major global concern. In this study, we leveraged an innovative CRISPR-pspCas13b technology and employed two key strategies to silence SARS-CoV-2 RNA and counteract its intrinsic ability to escape standard therapies through the generation of de novo mutations.

Firstly, the crRNA multiplexing approach to target various regions of viral RNA simultaneously mitigates the risk of target inaccessibility[47] and potential viral escape through genomic rearrangement or polymorphisms[7,8,16,48] (Fig. 4, Supplementary Figs. 4 and 7). Given the moderate mutation rate of SARS-CoV-2, it is highly unlikely that the virus would accumulate simultaneous mutations in various regions to escape a cocktail of 4 crRNAs without compromising its fitness. While the use of pooled crRNAs targeting various regions of the viral genome may not necessarily outperform a single crRNA in term of silencing efficiency, it provides a means to countervail the risk of viral escape through genomic rearrangements and polymorphisms. In fact, our data show that the NCP-1 crRNA achieved the highest silencing efficiency ($IC_{50} = 0.7$ pM) among all crRNAs in the NCP pool (Fig. 2), and combinations of various crRNAs did not further enhance viral suppression following viral infection (Fig. 4). It is worth noting that multiplexing highly effective crRNAs such as NCP-1 and SGP-2 is an appealing strategy to target various conserved subunits of the virus to simultaneously achieve high silencing while avoiding viral escape. This concept of multiplexed targeting is comparable to other combination therapeutic strategies that have proven to be effective against other viruses including HIV[49]. Compared to inhibitors of protein function that typically require years of modelling, design, and screening, the advantage of viral RNA targeting with pspCas13b lies in its design-flexibility, predictive efficacy and specificity, and the short time needed to generate the relevant crRNAs. The specificity, high silencing efficiency, and rapid deployment properties of pspCas13b that we have demonstrated are indicative of the translational potential of this technology to reshape the battle against SARS-CoV-2 including arising new variants.

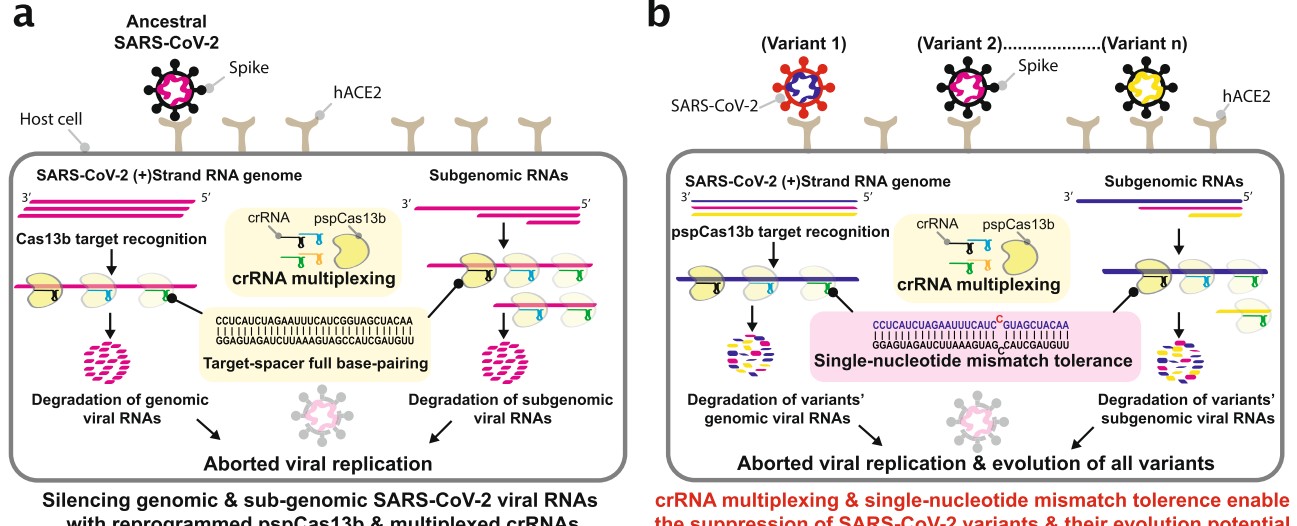

**Fig. 7 Schematic depiction of pspCas13b-mediated suppression of SARS-CoV-2 variants and their evolution through crRNA multiplexing and single-nucleotide mismatch tolerance. a** crRNA multiplexing enables simultaneous targeting of several genomic and sub-genomic RNA locations, which limits the probability of poor silencing due to target inaccessibility (e.g. RNA folding, interaction with RNA-binding proteins in vivo). The spacer of pspCas13b mediates sequence-specific target recognition through complete base-pairing with the targeted sequence, followed by nuclease activation and target degradation. The silencing of viral RNAs alters the integrity of the viral genome, suppresses sub-genomic RNA-dependent viral protein translation, and the assembly of replication-competent viral particles within infected mammalian cells. This viral RNA targeting approach aborts viral replication cycles and inhibits the infectivity of SARS-CoV-2 virus. **b** The error-prone RdRP creates viral polymorphisms (or variants) through the incorporation of single-nucleotide indels, and together with host (immune system) and environmental (antiviral therapies) positive pressures, result in the emergence of new strains with higher infectivity, pathogenicity, and therapy resistance. Unlike other antiviral therapeutics (e.g. antibodies-based approaches[7,8,11,17,18]), pspCas13b target affinity remains unaffected when a single-nucleotide indel occurs at the target site, enabling a single crRNA to be effective against wild type and mutant strains. Thus, pspCas13b mismatch tolerance and crRNA multiplexing approaches can suppress all variants and their viral evolution potential.

Likewise, other Cas13 orthologs[3,21,24,32,50,51] may also offer similar properties to control new RNA viruses and the emergence of new variants, although their mismatch tolerance in infection settings remains to be tested.

Secondly, our comprehensive mutagenesis analysis revealed that the positively charged central channel of pspCas13b can tolerate single-nucleotide mismatches within the RNA–RNA hybrid created by spacer-target base-pairing[30,31] (Figs. 1–3). We demonstrated that due to the mismatch tolerance described here, a single crRNA can simultaneously silence parental and SARS-CoV-2 variants (Figs. 3, 5–7). The mismatch tolerance mechanism shown here revealed that a single crRNA will likely remain effective against future variants that acquire de novo single-nucleotide mutations as a result of genome replication by the error-prone RdRP of SARS-CoV-2. Our mutagenesis data indicates that the spacer-target RNA-RNA duplex can tolerate between three to six nucleotide mismatches presumably depending on the spacer-target affinity and mismatch positions. Publicly available SARS-CoV-2 genome sequences[13,14] indicate that the probability of multiple concurrent mismatches within a single 30-nt region of SARS-CoV-2 is extremely unlikely, thereby suggesting that a single crRNA would remain effective against most variants. Targeting highly conserved genomic regions that are conserved in SARS-CoV-2 and other closely related coronaviruses is an interesting strategy to limit the potential viral escape from a single crRNA. Abbot et al.[26] bioinformatically predicted highly conserved genomic regions in coronaviruses and designed a pool of only six RfxCas13d crRNAs that should cover 90% of known coronaviruses.

We focused the in silico prediction of this study on the design of crRNAs targeting the SARS-CoV-2 RNA genome that were predicted to achieve high viral suppression efficiency via the

application of several molecular rules that should maximize crRNA transcription, folding, and target accessibility while avoiding mutation hotspots and potential off-targeting of human transcriptome (Supplementary Data files 1–5 & Supplementary Fig. 5). All together, these criteria are anticipated to maximize the overall pspCas13b silencing efficiency and provide useful tools to supress viral replication and interrogate the role of individual SARS-CoV-2 subunits in the life cycle of this new virus through loss-of-function assays.

Given the high degree of tolerance to single-nucleotide mismatch and the use of crRNA multiplexing (Fig. 7), we postulate that reprogrammed pspCas13b can act as a tool to silence SARS-CoV-2 strains that would escape conventional antiviral therapeutics, including recombinant monoclonal antibodies[7], antibodies in the plasma of convalescent patients, antibodies generated through vaccination[17,18], or small inhibitor molecules. A key step in enabling clinical translation of this proof-of-concept approach will be to develop a safe and effective delivery strategy such as lipid nanoparticle formulations for systemic, and possibly, mucosal delivery[52] for testing in animal models. Importantly, a CRISPR-Cas13 based viral suppression is also readily adaptable and expandable to other pathogenic viruses beyond SARS-CoV-2, and may therefore represent a powerful platform for antiviral therapeutics.

## Methods

**Design and cloning of pspCas13b guide RNAs.** Individual guide RNAs were cloned into the pC0043-pspCas13b[27] crRNA backbone (addgene#103854, a kind gift from Feng Zhang lab), which we refer to as crRNA backbone. This vector contains pspCas13b crRNA direct repeat (DR) sequence and two BbsI restriction sites for spacer cloning. A total of 20 μg of DNA backbone was digested by BbsI (NEB#) following the manufacturer's instructions (2 h at 37 C). The digested

backbone was gel purified using NucleoSpin™ Gel and PCR Clean-up Kit (Thermo Fisher 12), aliquoted, and stored in −20 °C.

For crRNA cloning, a forward and reverse single-stranded DNA oligos containing CACC and CAAC overhangs respectively, were ordered from Sigma (100 μM). A total of 1.5 μL of 100 μM the forward and reverse DNA oligos were annealed in 47 μL annealing buffer (5 μl NEB buffer 3.1 and 42 μL H₂O) by 5 min incubation at 95 °C and slow cooldown in the heating block overnight. 1 μL of the annealed oligos were ligated with 0.04 ng digested pspCas13b crRNA backbone in 10 μL of T4 ligation buffer (3 h, RT). All pspCas13b crRNA spacer sequences used in this study are listed in Supplementary Data file 8.

### Cloning of pspCas13b-NES-HIV-3xFlag-T2A-BFP.
The original pspCas13b (addgene#103862) is a gift from Feng Zhang lab[27]. A new pspCas13 plasmid was designed by fusing a 3xFlag-T2A-BFP tag to the pspCas13b C-terminus using NheI and EcoRI enzymatic restriction. A total of 5 μg of pUC19 vector encoding a psp-Cas13b-3′end-3xFlag-T2A-BFP sequence was generated by DNA synthesis (IDT). psp-Cas13b-3′end-3xFlag-T2A-BFP and the original pC0046-EF1a-pspCas13b-NES-HIV plasmids were digested by EcoRI/NheI restriction enzymes (1 h, 37 °C), followed by column clean up with NucleoSpin™ Gel and PCR Clean-up Kit (Macherey-Nagel). The digestion efficiency was verified by 1% agarose gel. The psp-Cas13b-3′end-3xFlag-T2A-BFP fragment with the correct size was gel purified using NucleoSpin™ Gel and PCR Clean-up Kit. The two fragments were ligated using T4 ligase (3 h, RT), and transformed into Stbl3 chemically competent bacteria. Positive clones were screened by PCR and Sanger sequenced (AGRF, AUS-TRALIA). The plasmid pspCas13b-NES-HIV-3xFlag-T2A-BFP is made available through Addgene (Ef1a-pspCas13b-NES-3xFLAG-T2A-BFP; #173029). The sequence is available in Supplementary Fig. 8.

### Cloning of SARS-CoV-2 Spike, Spike D614G, and nucleocapsid cDNA.
A mammalian expression plasmid containing a codon-optimized sequence of the Spike protein was obtained through Genscript plasmid sharing platform (Ref# MC_0101087), and was a kind gift from Dr. Haisheng Yu lab.

The coding sequence of NCP and part of Spike D614G sequence was designed according to the first SARS-CoV-2 genome[40]. The Genscript DNA synthesis platform provided the two sequences that were subsequently cloned into MSCV-IRES-mCherry vector in frame with 3xHA tag using BamHI digestion, gel purification, and ligation with T4 DNA ligase. The ligated product was transformed into chemically competent bacteria (Top10) and positive clones were screened by PCR and Singer sequencing (AGRF, AUSTRALIA). The NCP-3xHA-IRES-mcherry construct has been made available through Addgene (SARS-CoV-2 Nucleocapsid-3xHA-IRES-mcherry; #173030). The sequences of these DNA constructs are available in Supplementary Fig. 8.

### Plasmid amplification and purification.
TOP10 (for all crRNA, Spike, D614G Spike, and NCP cloning) and Stbl3 (for Cas13 cloning) bacteria were used for transformation. A total of 5–10 μL ligated plasmids were transformed into 30 μL of chemically competent bacteria by heat shock at 42 °C for 45 s, followed by 2 min on ice. The transformed bacteria were incubated in 500 μL LB broth media containing 75 μg/mL ampicillin for 1 h at 37 °C in a shaking incubator (200 rpm). The bacteria were pelleted by centrifugation at 6,010 rpm for 1 min at room temperature (RT), re-suspended in 100 μL of LB broth, and plated onto a pre-warmed 10 cm LB agar plate containing 75 μg/mL ampicillin, and incubated at 37 °C overnight. Next day, single colonies were picked and transferred into bacterial starter cultures and incubated for ~6 h for mini-prep or maxi-prep DNA purification according to the standard manufacturer's protocol. All crRNAs and pspCas13b clones that are generated in this study were verified by Sanger sequencing (AGRF, AUSTRALIA).

### Cell culture.
HEK 293 T (ATCC CRL-3216) and VERO (ATCC CCL-81) cell lines were cultured in DMEM high glucose media (Life Technologies) containing 10% heat-inactivated fetal bovine serum (Life Technologies), Penicillin/-Streptomycin, and -L-Glutamine (Life Technologies). Calu-3 cell line were cultured in advanced DMEM/F-12 media (Life Technologies) containing 10% heat-inactivated fetal bovine serum, Penicillin/-Streptomycin, and -L-Glutamine. HEK 293 T and VERO cells were maintained at confluency between 20 and 80% in 37 °C incubators with either 10% (HEK 293 T) or 5% (VERO/Calu-3). Cells were routinely tested and were mycoplasma negative.

### RNA silencing assays by transient transfection.
All transfection experiments were performed using an optimized Lipofectamine 3000 transfection protocol (Life Technologies, L3000015). For RNA silencing in HEK 293 T, cells were plated at approximately 30,000 cells/100 μL/96-well in tissue culture treated flat-bottom 96-well plates (Corning) 18 h prior to transfection. For each well, a total of 100 ng DNA plasmids (22 ng of pspCas13b-BFP construct, 22 ng crRNA plasmid, and 56 ng of the target gene) were mixed with 0.2 μL P3000 reagent in Opti-MEM Serum-free Medium (Life Technologies) to a total of 5 μL (mix1). Separately, 4.7 μL of Opti-MEM was mixed with 0.3 μL Lipofectamine 3000 (mix2). Mix1 and mix2 were added together and incubated for 20 min at room temperature, then 10 μL of transfection mixture was added to each well.

A similar protocol was used for VERO and Calu-3 cells transfection except 20,000 cells/well were seeded in a 96-well plate and transfected with a double volume of transfection mixture (20 μL). Supplementary Data file 9 summarizes the transfection conditions used in 96, 24, and 6-well plates for HEK 293 T, VERO, and Calu-3 cells.

After transfection, cells were incubated at 37 °C and 5% CO₂, and the transfection efficiency was monitored by either fluorescence microscopy or FACS analysis.

### Fluorescence microscopy analysis.
For RNA silencing experiments, the fluorescence intensity was monitored using EVOS M5000 FL Cell Imaging System (Thermo Fisher). Pictures were taken 48 h (HEK 293 T) and 72 h (VERO) post-transfection, and the fluorescence intensity of each image was quantified using a lab-written macro in ImageJ software. Briefly, all images obtained from a single experiment are simultaneously processed using a batch mode macro. First, images were converted to 8-bit, threshold adjusted, converted to black and white using Convert to Mask function, and fluorescence intensity per pixel measured using Analyze Particles function. Each single mean fluorescence intensity was obtained from four different field of views for each crRNA, and subsequently normalized to the non-targeting (NT) control crRNA. Two-fold or higher reduction in fluorescence intensity is considered as biologically relevant.

### Cell flow cytometry.
For monitoring cell transfection/transduction efficacy, cells were re-suspended in 200 μL 1x PBS containing 2% FBS for flow cytometry analysis. All samples were analyzed by an LSR II (BD Biosciences), FORTESSA X20 (BD Biosciences) or FACSymphony (BD Biosciences). All flow cytometry profiles were analyzed using FlowJo V10 software (Tree Star Inc).

### Western Blot.
Cells were washed three times with ice-cold PBS ± and lysed on ice in lysis buffer (50 Mm Tris, 150 mM NaCl, 1% NP-40, 2% SDS, pH 7.5). Samples were sonicated on low power, two-second pulses, and centrifuged at 16,000 g for 10 min, 4 °C. Supernatant was transferred to a new tube. Protein concentrations were quantified using the BCA assay (ThermoFisher Scientific) according to the manufacturer's instructions. A total of 10 μg proteins diluted in 1x Bolt LDS sample buffer and 1x Bolt sample reducing agent were denatured at 95 °C for 5 min. Samples were resolved by Bolt Bis-Tris Plus 4–12% gels in 1x MES SAS and transferred to PVDF membranes by a Trans-Blot Semi-Dry electrophoretic transfer cell (Bio-Rad) at 20 Volt for 30 min. Membranes were incubated in blocking buffer (5% (w/v) skin milk powder in PBST with 0.1% Tween 20) for 1 h at RT and probed overnight with primary antibody (anti-HA, mAb #2367, 1/2000 dilution, Cell Signaling Technology) at 4 °C. Blots were washed three times in PBST with 0.1% Tween20, followed by incubation with HRP-conjugated secondary antibody (Rabbit Anti-Mouse Immunoglobulins/HRP, 1/10000 dilution, #p0260, Dako) for 1 h at RT. Membranes were washed in PBST (0.1% Tween20) three times and incubated in appropriate strength ECL detection reagent. Chemiluminescence was detected using Invitrogen iBright Imaging Systems (Thermo Fisher Scientific).

### VERO and Calu-3 cells infection assays.
VERO and Calu-3 cells were transfected in 24-well plates and incubated for 48 h as described above. VERO cells expressing pspCas14b-BFP and various crRNAs were then infected in a level 3 containment laboratory with 200 μL of the ancestral, D614G, or the B.1.1.7 (known as the UK variant) SARS-CoV-2 virus isolate, a kind gift from Dr. Julian Druce[29] (Victoria Infectious Disease Reference Lab, VIDRL) at MOI of 0.1 or 0.01 in DMEM containing Penicillin/Streptomycin, L-Glutamine and 1 μg/mL TPCK-treated trypsin (LS003740, Worthington) to facilitate Spike cleavage and cell entry of the virus. After 1-h incubation at room temperature, virus production in supernatant was assessed by real-time PCR (RT-PCR) and infectivity assays at various time points. Calu-3 cells expressing pspCas13b-BFP and various crRNAs were infected with 200 μL of the ancestral virus isolate with 10⁴ TCID50/mL in DMEM containing 1% FBS, Penicillin/Streptomycin, L-Glutamine and 1 μg/mL TPCK-treated trypsin.

### RNA extraction, cDNA synthesis, and RT-PCR.
Viral RNA from 140 μL cell culture supernatant was extracted using the QIAamp Viral RNA Mini kit (#52906, Qiagen) following the manufacturer's instructions. SARS-CoV-2 RNA was converted to cDNA using the SensiFAST cDNA kit (#BIO-65053, BioLine) with 10 μL of RNA extract per reaction following the manufacturer's instructions. Quantitative RT-PCR reaction targeting RdRP gene was performed in triplicate in a Mx3005P QPCR System (Agilent) using PrecisionFAST qPCR Master Mix (PFAST-LR-1, Integrated Science). Total reaction mixture contains 2.5 μL cDNA, 0.75 μM forward primer (5′-AAA TTC TAT GGT GGT TGG CAC AAC ATG TT-3′), 0.75 μM reverse primer (5′-TAG GCA TAG CTC TRT CAC AYT T-3′) and 0.15 μM TaqMan Probe (5′-FAM-TGG GTT GGG ATT ATC-MGBNFQ-3′).

### Infectivity assays.
For titration of the 50% tissue culture infectious dose (TCID50) of SARS-CoV-2, VERO cells were plated in 96-well plates at 20,000 cells per well in MEM containing 5% heat-inactivated fetal bovine serum, Penicillin/Streptomycin, L-Glutamine, and 15 mM HEPES (Life Technologies). The cells were incubated overnight in a 5% CO₂ environment at 37 °C, washed once with PBS and then

cultured in serum-free MEM containing Penicillin/Streptomycin, L-Glutamine, 15 mM HEPES and 1 μg/mL TPCK-treated trypsin. A 10-fold initial dilution of samples with one freeze-thaw cycle was made in quadruplicate wells of the 96-well plates followed by 6 serial 10-fold dilutions. The last row served as negative control without addition of any sample. After a 4-day incubation, the plates were observed for the presence of cytopathogenic effect (CPE) using an inverted optical microscope. Any sign of CPE was categorized as a positive result. The endpoint titers were calculated by means of a simplified Reed & Muench method[53].

**Data analysis**. Data analyses and visualizations (graphs) were performed in GraphPad Prism software version 7. Specific statistical tests, numbers of independent biological replicates are mentioned in respective figure legends. The silencing efficiency of various crRNAs was analyzed using one-way Anova followed by Dunnett's multiple comparison test where we compare every mean to a control mean as indicated in the Figures (95% confidence interval). Unpaired two-tailed Student's t-test (95% confidence interval) was used to compare the viral titer between ancestral and D614G (Fig. 6). The P values (P) are indicated in the Figures. $P < 0.05$ is considered as statistically significant.

**Off-target predictions**. We used BWA (version 0.7.17-r1188) alignment tool[45] to align filtered crRNAs to human transcriptome (Homo_sapiens.GRCh38.cdna) to find potential off-targets. It runs in two steps, first one includes the alignment (bwa aln) by allowing various number of nucleotide mismatches, followed by mapping to transcriptomic coordinates (bwa samse). BWA aln was run with parameters taken from Aaron et al.[54].

**Reporting summary**. Further information on research design is available in the Nature Research Reporting Summary linked to this article.

## Data availability

All data are available in the main text and supplementary materials. Source Data are provided with this paper. All key plasmids constructed in this study, their sequences, and maps are deposited to Addgene (#173029–# 173030). Source data are provided with this paper.

## Code availability

The bioinformatic code for the design of single-nucleotide tiled crRNAs and spacer/target folding predictions are available here (https://github.com/data-vis/Covid19_crRNAs).

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

## Acknowledgements
We thank all lab members from the Trapani, Lewin, Voskoboinik, Subbarao, VIDRL, and Joo labs for facilitating experiments and discussions. We thank Dr. Georgia Deliyannis and Dr. David Jackson for providing the Calu-3 cells, and Dr. Julie McAuley for providing advices regarding Calu-3 cells infection. This work was supported by the National Health and Medical Research Council (NHMRC) of Australia through a program grant to J.A.T., and a program grant and practitioner fellowship to S.R.L. M.F. is supported by a Peter MacCallum Cancer centre strategic plan funding in partnership with the Childhood Cancer Institute Australia (CCIA). M.F., P.G.E., and J.A.T. are supported by a Cancer Council Victoria Ventures grant (ID 829606).

## Author contributions
M.F. conceived the study. M.F., J.A.T., and S.R.L. supervised the study. M.F., J.A.T., S.R.L., W.Z., W.H., and J.M.L.C. designed the experiments and discussed all data. M.F., W.H., and J.M.L.C. cloned all constructs, optimized and performed virus-free silencing assays, and analysed the data. W.Z. performed all SARS-CoV-2 infection assays with help from R. R., J.M.Z., and D.F. W.Z. analysed data from infectivity assays. M.F. and W.Z. extracted viral RNAs, performed RT-PCR, and analysed the data. A.K. performed the computational analysis of SARS-CoV-2 guide and target RNAs, mutation hotspots, and predicted off-targeting with input from M.F. I.V. P.G.E., D.F.G.P., and J.S. discussed the project and the data. J.M.Z. provided access to Calu-3 cell line and discussed cell culture conditions. M.F. generated all graphs, figures, and wrote the manuscript. M.F., J.A.T., and S.R.L. revised and edited the manuscript. All authors read, commented, edited, and approved the manuscript.

## Competing interests
S.R.L. is a member of advisory boards of Merck and Gilead and has received investigator-initiated industry funded grants from Merck, Gilead, and Viiv. None of this support is relevant to the work in this manuscript. The remaining authors declare no conflict of interest.
