## [Peer Review File · Nature Communications]

Reviewers' Comments:

Reviewer #1:

Remarks to the Author:

Fareh and colleagues strongly demonstrate that PspCas13b is able to target SARS-CoV-2 RNA using virus-free, fluorescent reporter systems and using a complete cell culture model of SARS-CoV-2 infection in Vero cells. Numerous PspCas13-crRNAs, designed against the spike and the nucleoprotein and other open reading frames, can effectively silence viral gene expression and can reduce replication. The authors also characterized the effects of mutations within the target and crRNA binding site which is particularly important in the context of viral evolution, highlighting sustained silencing activity even in the context of Spike D614G which is a critical example of SARS-CoV-2 evolution. The paper is well written and easy to follow.

In signing this review (as Dr. Catherine A. Freije), I want to reiterate that I believe this manuscript would be a fantastic addition to the growing collection of work demonstrating of how CRISPR-Cas13 could represent a new strategy for responding to the current COVID-19 pandemic and in the future. My major and minor concerns are geared to make the authors' results and conclusions even stronger and the work an even greater resource for the community,

Major concerns:

1. Because it is argued that that the bioinformatic analysis of all potential SARS-CoV-2 PspCas13b crRNAs is a valuable investigative tool, additional information and other considerations should be incorporated into this analysis and the crRNA set. The following information could be included (1) the coverage or number of crRNAs for each SARS-CoV-2 open reading frame, (2) how many crRNAs in the final set target both the genomic and subgenomic RNAs, (3) what is the viral diversity across these targets based on publicly available sequence data (for example these crRNAs might not be good targets if they lie in a site with high levels of diversity), (4) are there any that could be pan-human coronavirus targets? Furthermore, experimental evidence should be provided that justifies the filtering strategy by testing the silencing activity of crRNAs with or without polyT stretches and with or without RNA-RNA duplexes. It is also possible that the secondary structure filtering is too stringent, as a previous study identified that G-quadruplexes in particular show the largest effect of reduced target knockdown (Wessels et al. 2020 Nature Biotechnology doi: 10.1038/s41587-020-0456-9)

2. The statement in lines 137-138 that the degree of variation in base pairing that is tolerated for PspCas13b is not known is incorrect. All single and double mismatches across a target were tested in Cox et al. 2017 Science doi: 10.1126/science.aag0180 (Figure S3). The testing done in this paper not identical and still valuable, but the results here should be compared to the results in Cox et al. particularly discussing what could drive any differences between these two datasets.

3. The data and interpretation of the results of Figure 1G should be revisited because this data is an N of 1. This data should also incorporate some of the bioinformatic analysis used to filter SARS-CoV-2 crRNAs at the start of the manuscript. How is crRNA folding affected over the course of this tiling? Are there polyTs that could explain the data? Wessels et al. all references other factors that could also be influence the variation in silencing observed – strong crRNAs have an enrichment of Cs nearby the target and enrichment of the preferred cleavage base upstream of the target. Any of these factors could be alternative explanations of the clustering observed and mentioned in line 164.

4. Although pooling crRNAs has been shown previously to increase the effectiveness of Cas13 targeting, this claim is made here (lines 278-284) without sufficient evidence from the results presented in Figure 4D-F. The individual performance of each crRNA in the pool is not presented. Therefore, it is unclear if the pool is effective because of all 4 crRNAs present or if one (or more) strongly-performing crRNA is driving the observed effect.

5. A few additional things should be considered in the interpretation and experiments surrounding the targeting of the Spike D614G region. This mutation is a A-to-G change which means that the crRNA containing a U at the mutated site will still bind due to wobble-base pairing. Although, this is beneficial for the purposes of using Cas13 to target this variant, non-A-to-G mutations could

have a different outcome. Therefore, it would be useful to test the effect of when the crRNA contains a base at that site that does not wobble base pair to extend the conclusions more broadly. Furthermore, it is worth commenting that even though Cas13 is effective despite the mutation in the crRNA-target site that the effect sizes are not as pronounced, yet this might be hard to decouple from the increased infective potential of the D614G variant since the NCP-1 positive control also appears to be a bit less effective.

Minor concerns:

1. It is more common in the Cas13 literature to define the RNA sequence that programs Cas13 to its target the CRISPR RNA (crRNA) as opposed to the gRNA in order to not confuse this RNA with that of the tracrRNA-containing gRNA of Cas9.

2. In line 167-169, it is mentioned that the single nucleotide tiling data suggest that activity is independent of a specific PFS. I would not interpret this data as definitively because the tiling changes more than just the PFS (crRNA secondary structure, accessibility of the target site, flanking nucleotide content, etc.) and previous studies have tested in this in a more controlled manner although in bacteria (Cox et al., Figure S1).

3. The results presented in Figure 1B and 2B would be better visualized in color like they were presented in Figure 1A and 2A, respectively.

4. Lines 215 and 398, change indel to mutation or single-nucleotide variant.

5. Line 252, provide a quantitative result for the fold increase in viral RNA or the reduction from the non-targeting control to remain consistent with the quantitative reporting of Cas13's effects as was done in other portions of the text.

6. In many figure y-axes, change A.U to A.U.

7. Figure 6 is not referenced in the main text.

8. Please clarify in the figure captions what each data point represents (e.g. 1 image field of view in a biological replicate) and include data points when $N > 1$ (e.g. Figure 4B-E)

Reviewer #2:

Remarks to the Author:

This report is of interest to the international scientific and clinical community as it provides a novel approach to SARS-CoV-2 therapeutics that has not been previously developed. This is especially noteworthy as it demonstrates efficacy following specific viral mutations of interest. The work extends prior genome-wide screens that were used to identify host therapeutic targets. such targets were independent of the virus itself and included kinase SRPK1 and SRPK2 that were proven essential for SARS-CoV-2 replication as inhibition of both with small molecules led to a dramatic decreases in virus production. There is very limited parallel information in the published literature outlining direct viral excision through CRISPR-Cas systems.

There are points of interest and innovation and others that serve to detract from the impact of the work. My opinion in finality is that the authors should be given the opportunity to correct the deficiencies and have the paper re-submitted once that is completed.

The points that raise immediate interest are as follows.

First it is clear that mutation-driven evolution of SARS coronavirus-2 (SARS-CoV-2) underlies the immediate need for the development of innovative approaches that can both suppress viral growth and circumvent viral escape. The work contained in this report has the potential to do both but the specificity, efficiency and off-target toxicities are incomplete in development and discussion.

Second, the use of genome-wide computational prediction and single nucleotide resolution

screening to reprogram CRISPR-Cas13b against SARS-CoV-2 genomic and subgenomic RNAs while innovative descriptions for alternatives and efficiency require better discussion.

Third, the demonstration that reprogrammed Cas13b that are able to target subgenomic spike and nucleocapsid viral transcripts and silence virus is noteworthy.

Fourth the results were optimized by the generated gRNA suppression of viral replication of > 90% in mammalian cells infected with replication-competent SARS-CoV-2. This provides early proof of concept but the efficiency of the transfection system deployed are not discussed.

Fifth, the likely most exciting part of this study rests in the results of the single-nucleotide mismatch which showed that it did not impair the capacity of a single gRNA to suppress ancestral and mutated SARS-CoV-2. Other potential nucleotide mismatches should be discussed or developed in the report.

Specific and notable limitations in extrapolating or at the least supporting the data sets found in this report to any "real life" virus infected person. These are as follows.

First, all of the virological data is from transfection of recombinant CRISPR gRNAs into infected virus target cells. The transfections are limited to subpopulations of cells and may not be the same as those that are infected with virus.

Second, the transfected cells are quite distinct from transduced cells that would enter the cells through natural routes of delivery. Thus there need be included in this report a natural delivery system for the CRISPR-Cas that could be lipid nanoparticles, lentivirus or adenovirus as each represent common systems for delivery of such therapeutics.

Third, VERO cells or any of the cell lines employed are not natural SARS-CoV-2 targets so replicate experiments on viral inhibition/silencing need be performed on epithelial cells or other targets relevant to natural infection.

All together the demonstration of maximal viral suppression in artificial virus free-models or transfected cells is not yet biologically sufficient to demonstrate therapeutic efficacy.

Manuscript No:

Dear Dr. Cloney,

We thank you for giving us the opportunity to revise our work, and the reviewers for their enthusiasm and constructive comments. We have revised our manuscript in response to the reviewers' comments and have performed additional experiments, analyses, and restructured the manuscript. Below, we offer a point by point response to each issue raised by the reviewers.

Reviewer #1 (Remarks to the Author).

Fareh and colleagues strongly demonstrate that PspCas13b is able to target SARS-CoV-2 RNA using virus-free, fluorescent reporter systems and using a complete cell culture model of SARS-CoV-2 infection in Vero cells. Numerous PspCas13-crRNAs, designed against the spike and the nucleoprotein and other open reading frames, can effectively silence viral gene expression and can reduce replication. The authors also characterized the effects of mutations within the target and crRNA binding site which is particularly important in the context of viral evolution, highlighting sustained silencing activity even in the context of Spike D614G which is a critical example of SARS-CoV-2 evolution. The paper is well written and easy to follow. In signing this review (as Dr. Catherine A. Freije), I want to reiterate that I believe this manuscript would be a fantastic addition to the growing collection of work demonstrating of how CRISPR-Cas13 could represent a new strategy for responding to the current COVID-19 pandemic and in the future. My major and minor concerns are geared to make the authors' results and conclusions even stronger and the work an even greater resource for the community,

We thank the reviewer#1 for her enthusiasm and the positive feedback. We have revised the manuscript to which now includes additional bioinformatic analysis and the results of wet lab experiments suggested, expanded the discussion and restructured the paper. We feel addressing these comments has greatly improved the quality of this manuscript. Below is our point-by-point reply to the comments.

Major concerns:

1. Because it is argued that that the bioinformatic analysis of all potential SARS-CoV-2 PspCas13b crRNAs is a valuable investigative tool, additional information and other considerations should be incorporated into this analysis and the crRNA set. The following Information could be included (1) the coverage or number of crRNAs for each SARS-CoV-2 open reading frame, (2) how many crRNAs in the final set target both the genomic and subgenomic RNAs, (3) what is the viral diversity across these targets based on publicly available sequence data (for example these crRNAs might not be good targets if they lie in a site with high levels of diversity), (4) are there any that could be pan-human coronavirus targets? Furthermore, experimental evidence should be provided that justifies the filtering strategy by testing the silencing activity of crRNAs with or without polyT stretches and with or without RNA-RNA duplexes. It is also possible that the secondary structure filtering is too stringent, as a previous study identified that G-quadruplexes in particular show the largest effect of reduced target knockdown (Wessels et al. 2020 Nature Biotechnology doi: 10.1038/s41587-020-0456-9).

- (1) *We have used publicly available SARS-CoV-2 genome annotation (UCSC genome browser) to map each crRNA to a specific genomic location. We added a new column in the **Supplementary Tables 2, 3, & 4** to highlight crRNAs that exclusively target subgenomic regions and the ones that target intergenic regions. We also provide a **Supplementary Figure 7** generated in UCSC genome browser where we aligned the top 838 crRNAs to SARS-CoV-2 genome. We updated the manuscript with this information and indicated in the discussion that targeting subgenomic regions may yield better efficacy due to the high abundance of*

targets and the high probability of encounter between *pspCas13b* and its target transcripts either as RNA positive-strand genome or mRNA.

- (2) In the top hit list (**Supplementary Table 3**), we found 827 crRNAs targeting coding regions and only 11 targeting intergenic regions that are not transcribed. Among the crRNAs targeting open reading frames, we found 578 targeting ORF1ab, 148 targeting the Spike (S), 25 targeting ORF3a, 4 targeting the envelop (E), 22 targeting M, 8 targeting ORF6, 4 targeting ORF7a, 30 targeting ORF7b, 4 targeting ORF8, and 4 targeting N. We provide a table that summarizes this new analysis (**Supplementary Table 3**) and discussed the data in the new version of the manuscript.
- (3) We used publicly available genomic data from GISAID and the UCSC genome browser to highlight mutation hotspots on the genome of SARS-CoV-2. There are 18 hotspots that are frequently mutated (0.04 mutation frequency, Gosmawi et al, 2021) genome wide. We highlighted in red the crRNAs that target these mutation hotspots in **Supplementary Tables 3 & 4**. Of note, there are only 21 and 10 crRNAs that overlap with the mutation hotspots in **Supplementary Tables 3 & 4**, respectively. We highlighted these crRNAs in red in the tables. We discuss this new information in the revised manuscript, where we recommend not to use crRNAs that target mutation hotspots due to an increased risk of mismatch.
- (4) A list of pan-human crRNAs that target all human coronaviruses has been addressed in a previous manuscript (Abbot et al, Cell, 2020). To avoid redundancy with this study and keep the focus on SARS-CoV-2, we generated a list of crRNAs (**Supplementary Tables 3 & 4**) that target genomic regions of SARS-CoV-2 that are conserved and should be effective against all the reported SARS-CoV-2 strains. Additionally, the use of a cocktail of crRNAs should mitigate the risk of mutational escape. We further discuss this point in the new manuscript.

Filtering Strategy:

We designed additional crRNAs harbouring 4T or 5T repeats at various positions to target NCP in our reporter model. Due to the polyT sequence, the yield of U6 promoter transcription was anticipated to be reduced. Our new experimental data shows that including a polyT sequence within the spacer leads to poor silencing efficiency compared to crRNAs without polyT sequences (Supplementary Figure 6A & 6B). All crRNAs harbouring 5T yielded poor silencing efficiency (<~50%) (Supplementary Figure 6A). Also, of 8 crRNAs that included a 4T, 6 yielded a very poor silencing efficiency (Supplementary Figure 6B). Overall, this new data shows that targeting polyT sequences with *pspCas13b* decreases the likelihood of silencing - likely due to premature transcription termination (Zongliang Gao et al, *Mol Ther Nucleic Acids*, 2018). We discuss this new data in the manuscript.

We also tested the impact of including a G-quadruplex structure at various locations in the target using the NCP-mcherry reporter model. 6 crRNAs out of 9 with a 4C in the spacer showed poor silencing efficiency, while the other 3 crRNAs exhibited moderate to good silencing efficiency. Overall, the data indicates that indeed, the presence of G-quadruplex sequences in the target may reduce the likelihood of silencing with *pspCas13b* (Supplementary Figure 6C). We updated the manuscript with this new data.

2.The statement in lines 137-138 that the degree of variation in base pairing that is tolerated for *PspCas13b* is not known is incorrect. All single and double mismatches across a target were tested in Cox et al. 2017 Science doi: 10.1126/science.aaq0180 (Figure S3). The testing done in this paper not identical and still valuable, but the results here should be compared to the results in Cox et al. particularly discussing what could drive any differences between these two datasets.

We thank the reviewer for drawing attention to our inaccurate statement. We revised the statement in the manuscript and further acknowledged the mutagenesis work performed by Cox et al. Cox et al conducted single and double nucleotide mutagenesis in bacteria using library plasmids, while we investigated 3, 6, 9, 12, 15, 18, 21, 24, 27, and 30 nucleotide mismatches at various positions in mammalian cells using individually cloned single crRNA. Our mutagenesis data in both Figure 1F & 2G-I conducted with two different spacers indicate that a 3-nt mismatch at the central region 14-16 were tolerated, while Cox et al observed that the central region between 12-26 was sensitive to mismatches. We feel these two investigations are very complementary and the discrepancy in the silencing efficiency with mismatches at position 14-16 could be due to the difference in the experimental settings (library vs single cloned gRNAs), and the organisms studied (bacteria vs mammalian cells). We also noticed that the impact of mismatches on silencing is dependent on the spacer sequence probed. Mismatches within crRNA2 targeting spike had a higher impact on the silencing efficacy compared to the crRNA1 targeting NCP. We speculate that spacers with

Supplementary Figure 6

higher target affinity are likely to tolerate mismatches more readily than spacers with a lower affinity. We revised the result and discussion sections to reflect this important point.

We are cautious not to over-emphasise differences from Cox et al, in our mutagenesis data since the differences in outcome may simply reflect differences in the spacer sequences. We believe a dedicated investigation focused on correlating spacer mismatch tolerance with silencing efficiency is needed to comprehensively uncover the molecular basis of *pspCas13b* targeting and to develop targeting 'rules' that will help guide the choice of the most ideal spacer sequences. We are currently conducting such investigation using individually cloned crRNAs as well as pooled library screens that should address this point, and plan to publish the findings in a separate paper.

3. The data and interpretation of the results of Figure 1G should be revisited because this data is an N of 1. This data should also incorporate some of the bioinformatic analysis used to filter SARS-CoV-2 crRNAs at the start of the manuscript. How is crRNA folding affected over the course of this tiling? Are there polyTs that could explain the data? Wessels et al. all references other factors that could also be influence the variation in silencing observed – strong crRNAs have an enrichment of Cs nearby the target and enrichment of the preferred cleavage base upstream of the target. Any of these factors could be alternative explanations of the clustering observed and mentioned in line 164.

To address the reviewer's comment and increase the confidence in the data in Figure 1G we performed an additional experiment and now present the data as N=2. The silencing profile remained similar to the one we reported in the previous version. Below is the data from the two experiments, presented separately. We also added a schematic showing the Spike RNA sequence covered by the tiled crRNAs to further highlight sequence composition of target and spacers used (Fig. 1G, Supplementary Fig. 1).

Replicate 1 (Fig. 1G)

Replicate 2 (Suppl. Fig. 1)

We re-analyzed the nucleotide composition of tiled crRNAs in Fig. 1G and ruled out the presence of polyT sequence with more than 3 successive T's among the crRNAs tested that could prematurely terminate transcription. We found that 20 tiled crRNAs (positions 1-20) contained a G-quadruplex (at various spacer positions) but this did not appear to compromise silencing efficiency as the majority achieved silencing efficiency higher than 75%. We discuss this new information in the current version of the paper.

We acknowledge that Wessels et al used ~24,000 crRNAs in library screen assays combined with machine learning approach to comprehensively investigate the targeting rules of RfxCas13d ortholog in a dedicated study. However, the targeting rules for pspCas13b might be quite different from RfxCas13d as the two proteins have low sequence homology. The aim of the experiment presented in Fig. 1G was to provide a single-nucleotide resolution analysis of how silencing efficiency varies spatially across a given target RNA sequence. Given the silencing efficiency is governed by multitude of parameters (e.g. RNA target accessibility, crRNA folding, spacer nucleotide composition), we feel the size of tiled crRNAs used in the Fig. 1G is too small to comprehensively investigate the rules governing pspCas13b silencing. We are also currently investigating pspCas13b targeting rules using individually cloned crRNAs and a library screen-based approach containing several thousands of crRNAs combined with machine-learning and plan to publish the results separately.

4. Although pooling crRNAs has been shown previously to increase the effectiveness of Cas13 targeting, this claim is made here (lines 278-284) without sufficient evidence from the results presented in Figure 4D-F. The individual performance of each crRNA in the pool is not presented. Therefore, it is unclear if the pool is effective because of all 4 crRNAs present or if one (or more) strongly-performing crRNA is driving the observed effect.

We tested individual crRNAs in the Spike and NCP pool in the virus-free reporter system and we show that they have various silencing efficiency (Fig. 1&2). In the infection model, we show that both NCP1 (single crRNA) and a pool of NCP-targeting crRNAs can achieve high silencing efficiency (Fig. 4). The use of pooled crRNAs is a conservative approach to avoid viral escape in the unlikely event of several mutations occurring within a targeted sequence of a single crRNA (e.g. deletion or substitution of several bases). We clarified this point in the revised manuscript.

5. A few additional things should be considered in the interpretation and experiments surrounding the targeting of the Spike D614G region. This mutation is a A-to-G change which means that the crRNA containing a U at the mutated site will still bind due to wobble-base pairing. Although, this is beneficial for the purposes of using Cas13 to target this variant, non-A-to-G mutations could have a different outcome. Therefore, it would be useful to test the effect of when the crRNA contains a base at that site that does not wobble base pair to extend the conclusions more broadly. Furthermore, it is worth commenting that even though Cas13 is effective despite the mutation in the crRNA-target site that the effect sizes are not as pronounced, yet this might be hard to decouple from the increased infective potential of the D614G variant since the NCP-1 positive control also appears to be a bit less effective.

To address this point, we designed and tested 14 additional crRNAs targeting the Spike D614G region. We tested side-by-side crRNAs with a C (to create a full match with the D614G target) or with either a G or U as a single-nucleotide mismatch at spacer positions 5, 10, 15, 20, 25, and 30. Overall, U or G at the mismatch position did not dramatically change the landscape of silencing as a single-mismatch per spacer is well tolerated. The new data shows a moderate reduction in silencing efficiency when a G was introduced at position 15 or 30. Conversely, a G mismatch at position 5 improved the silencing efficiency compared to U or C at the same position (Fig. 5A, 5B, & 5C). We include this data into the new version of the manuscript in a separate Figure and we discuss it in the revised manuscript.

Figure 5

Minor concerns:

1. It is more common in the Cas13 literature to define the RNA sequence that programs Cas13 to its target the CRISPR RNA (crRNA) as opposed to the gRNA in order to not confuse this RNA with that of the tracrRNA-containing gRNA of Cas9.

We replaced gRNA with crRNA throughout the manuscript.

2. In line 167-169, it is mentioned that the single nucleotide tiling data suggest that activity is independent of a specific PFS. I would not interpret this data as definitively because the tiling changes more than just the PFS (crRNA secondary structure, accessibility of the target site, flanking nucleotide content, etc.) and previous studies have tested in this in a more controlled manner although in bacteria (Cox et al., Figure S1).

We apologise for the confusion here. We meant that our data suggest that the PFS doesn't dominate the likelihood of achieving silencing efficiency. For instance, in the tiling experiment (Fig 1G) crRNAs between positions 15 and 29 achieved high silencing efficiency (>75%) despite the diversity in the PSF generated by the single-nucleotide tiling. We think that the discrepancy with Cox et al may be due to the approach (library screen versus cloned single crRNAs) and the model organism (bacteria versus mammalian cells) used. We acknowledge that although individual crRNAs allowed us to probe only a small region of RNA (61 nucleotides long), this approach had higher penetrance than a cas13 library screening as it was not limited by delivery of a single crRNA copy per cell. We demonstrated in the dose-dependent silencing experiments (Fig. 1&2) that crRNA delivery and expression in the cell was key to efficient silencing. Therefore, we have high confidence in the findings of our tiling experiment. We tuned down the PSF statement to "data suggests" and further discuss these points in the revised manuscript.

3. The results presented in Figure 1B and 2B would be better visualized in color like they were presented in Figure 1A and 2A, respectively.

*We changed the color and the overall workflow chart in both **Fig. 1** and **Fig. 2** to facilitate understanding of our assays.*

Figure 1

Figure 2

4. Lines 215 and 398, change indel to mutation or single-nucleotide variant.
We changed indel to single-nucleotide variant.

5. Line 252, provide a quantitative result for the fold increase in viral RNA or the reduction from the non-targeting control to remain consistent with the quantitative reporting of Cas13's effects as was done in other portions of the text.

Now we provide a quantitative result for these data as suggested.

6. In many figure y-axes, change A.U to A.U.

We changed A.U to A.U.

7. Figure 6 is not referenced in the main text.

*Now we refer to **Fig. 6** in the manuscript.*

8. Please clarify in the figure captions what each data point represents (e.g. 1 image field of view in a biological replicate) and include data points when N>1 (e.g. Figure 4B-E).

We have now further clarified the figure caption as suggested and included datapoints in graphs when N>1.

Reviewer #2 (Remarks to the Author):

This report is of interest to the international scientific and clinical community as it provides a novel approach to SARS-CoV-2 therapeutics that has not been previously developed. This is especially noteworthy as it demonstrates efficacy following specific viral mutations of interest. The work extends prior genome-wide screens that were used to identify host therapeutic targets. Such targets were independent of the virus itself and included kinase SRPK1 and SRPK2 that were proven essential for SARS-CoV-2 replication as inhibition of both with small molecules led to a dramatic decrease in virus production. There is very limited parallel information in the published literature outlining direct viral excision through CRISPR-Cas systems.

There are points of interest and innovation and others that serve to detract from the impact of the work. My opinion in finality is that the authors should be given the opportunity to correct the deficiencies and have the paper re-submitted once that is completed.

The points that raise immediate interest are as follows.

First it is clear that mutation-driven evolution of SARS coronavirus-2 (SARS-CoV-2) underlies the immediate need for the development of innovative approaches that can both suppress viral growth and circumvent viral escape. The work contained in this report has the potential to do both but the specificity, efficiency and off-target toxicities are incomplete in development and discussion.

We thank the reviewer#2 for the positive and valuable feedback. We believe that addressing the comments raised significantly improved the quality of this manuscript.

To address the concern of off-target activity, we performed additional bioinformatic analyses to predict any potential off-target activity across the transcriptome using the human transcriptome (GRCh38.cdna) as a reference. We used Burrows-Wheeler Alignment tool (BWA) and allowed for various numbers of mismatches between the spacer and target. In the top hit list we found no crRNA that would base pair with any human transcript, even when 1 or 2 nucleotide mismatches are allowed. Off-target activity with 3, 4, 5 or 6 nucleotide mismatches were found in 6, 21, 61, and 344 crRNAs, respectively (Supplementary Table 5). To accommodate these findings, we took a very conservative approach and excluded any crRNA predicted to have 6 or fewer nucleotide mismatches with any human transcript. This stringent filtering strategy identified 344 crRNAs in the top hit list, and reduced the number of crRNAs from 838 to 494 (Supplementary table 4). It is worth noting that due to the uniquely long (30-nt) spacer of pspCas13b we can permit such stringent filtering to theoretically eliminate the likelihood of off-target activity. Accordingly, Cas13 orthologs with shorter spacer sequences may have higher off-target activity rates as the specificity is proportional to spacer-target basepairing. We acknowledge that this bioinformatic prediction requires experimental validation with whole transcriptome sequencing to evaluate off-target activity with various crRNAs in viral infection and silencing settings. We have provided additional tables (Supplementary table 4 & 5) with this off-target activity prediction data and discuss the information in the revised manuscript.

As for efficiency, both virus-free and viral infection models demonstrate high silencing efficiency reaching up to 99% and 90% target silencing, respectively. We further highlighted this finding in the discussion.

Second, the use of genome-wide computational prediction and single nucleotide resolution screening to reprogram CRISPR-Cas13b against SARS-CoV-2 genomic and subgenomic RNAs while innovative descriptions for alternatives and efficiency require better discussion.

We acknowledge that the computational prediction was brief and could have been more detailed, so we revisited this section. We have now:

- (i) Provided additional experimental data supporting filtering strategies such as excluding spacers with poly-T or poly-C sequences that might impair silencing efficiency and discussed these new findings in the manuscript (**Supplementary Fig. 6**).
- (ii) Mapped each crRNA in the top hit list to the SARS-CoV-2 genome and highlighted the targeted open reading frames (ORF) and intergenic regions (**Supplementary Tables 3 & 4**).
- (iii) Conducted new bioinformatic analyses to highlight crRNAs targeting SARS-CoV-2 genomic locations (in red) with very frequent SARS-CoV-2 mutation hotspots and discussed the risk of viral escape due to occurrence of multiple mismatches (**Supplementary Tables 3 & 4**).
- (iv) Conducted a human transcriptome-wide computational prediction to exclude crRNAs with potential off-targeting (**Supplementary Tables 4 & 5**).
- (v) Restructured the result section by moving the computational prediction to the end. We believe this improves the flow and the overall structure of the paper.
- (vi) Discuss the new information and usefulness of the computational analysis in the updated discussion (also see point-by-point response to reviewer#1 above).

Third, the demonstration that reprogrammed Cas13b that are able to target subgenomic spike and nucleocapsid viral transcripts and silence virus is noteworthy.

We thank the reviewer for this comment. We further highlighted in the discussion the advantage of targeting subgenomic spike and nucleocapsid transcripts for higher suppression efficiency and to mitigate the risk of viral escape.

Fourth the results were optimized by the generated gRNA suppression of viral replication of > 90% in mammalian cells infected with replication-competent SARS-CoV-2. This provides early proof of concept but the efficiency of the transfection system deployed are not discussed.

In the revised version we further discuss how transfection efficiency (as measured by microscopy and FACS) can have an impact on viral replication in infected cells. While analytical assays examining BFP fluorescence show modest transfection efficiency (25-40%), we were nonetheless able to suppress viral titre by 80-90%. The high degree of viral suppression suggests we may be underestimating the efficiency of Cas13 delivery due either to Cas13-BFP fluorescence being below the detection threshold or BFP protein having reduced stability compared with Cas13. We further discuss this point in the revised manuscript.

Fifth, the likely most exciting part of this study rests in the results of the single-nucleotide mismatch which showed that it did not impair the capacity of a single gRNA to suppress ancestral and mutated SARS-CoV-2. Other potential nucleotide mismatches should be discussed or developed in the report.

- (i) We thank the reviewer for the enthusiasm regarding our concept of resilience to viral escape through single-mismatch tolerance. We are thrilled they found this original idea exciting, which is the main focus of this paper. We further discuss the implication of this concept in the new version of the paper and the need to validate it against other SARS-CoV-2 variants.
- (ii) To consolidate the single-nucleotide tolerance data while addressing the comment of reviewer#1, we designed 14 additional crRNAs targeting the D614G mutation that created either a full match with the target when a C base is used, or G and U as a mismatch at various position of the spacer. Overall, we show that single-nucleotide mismatch is well-tolerated, and the type of nucleotide at the mismatch position has a very limited or no impact on the silencing (**Fig. 5**)
- (iii) Of note, in the revised manuscript we provide additional data that show crRNA1 targeting the Nucleocapsid (NCP1) also efficiently suppressed replication-competent **UK variant B.1.1.7** in both RT-PCR and infectivity assays (**Fig. 4H**). The idea here was that

designing crRNA that target well-conserved and accessible genomic regions of SARS-CoV-2 can lead to efficient suppression of strains that may become resistant to other therapeutic or immune interventions. We discuss this new data and concept in the revised paper. We are very excited about including this additional new data, as the B.1.1.7 variant is now the dominant global variant of concern but had not emerged at the time we originally submitted our manuscript.

Specific and notable limitations in extrapolating or at the least supporting the data sets found in this report to any "real life" virus infected person. These are as follows.

First, all of the virological data is from transfection of recombinant CRISPR gRNAs into infected virus target cells. The transfections are limited to subpopulations of cells and may not be the same as those that are infected with virus.

We thank the reviewer for this important remark. We believe that the transfection efficiency is under-estimated for the reasons listed above. Additionally, silencing the virus in only a subpopulation of cells may mean that the virus can secondarily infect cells that are untransfected and so do not express Cas13 and crRNA. We acknowledge that additional investigation using integrated lentiviruses, adenoviruses, and lipid nanoparticles with near 100% delivery efficiency and by mimicking "real life" infection settings is needed and is anticipated to achieve near complete and sustainable viral suppression.

Second, the transfected cells are quite distinct from transduced cells that would enter the cells through natural routes of delivery. Thus, there need be included in this report a natural delivery system for the CRISPR-Cas that could be lipid nanoparticles, lentivirus or adenovirus as each represent common systems for delivery of such therapeutics.

We acknowledge that alternative delivery methods will be needed to develop viable in vivo therapies. Our work to this point is a proof-of-concept study showing that we can target replication competent SARS-CoV-2 in mammalian cells and that our methodology may be capable of countering viral evolution through mismatch tolerance. The methods used here have been optimized to achieve this goal in cells that can be transfected. We are aware that the development of lipid nanoparticles for in vivo delivery, and the subcloning of pspCas13b and crRNAs for AAVs or lentiviral packaging will require several months of work and are thus beyond the scope of the current study. Rather, we believe that sharing our findings will allow experts in nanodelivery to build on our findings and test a variety of approaches to limit SARS-CoV-2 replication in model organisms. We discuss these points in the new version of the manuscript.

Third, VERO cells or any of the cell lines employed are not natural SARS-CoV-2 targets so replicate experiments on viral inhibition/silencing need be performed on epithelial cells or other targets relevant to natural infection.

In addition to VERO cells, we now show that PspCas13b and a crRNA targeting NCP can silence SARS-CoV-2 in the human lung epithelial cell line Calu-3 (Fig. 4G).

Altogether, the demonstration of maximal viral suppression in artificial virus free-models or transfected cells is not yet biologically sufficient to demonstrate therapeutic efficacy.

We have revised the discussion to reflect this point and to further highlight that this is a proof-of-concept study that demonstrates the molecular basis of silencing at the single-nucleotide level. We also discuss the need for in vivo investigation to demonstrate therapeutic efficacy in animal models using various delivery approaches.

Reviewers' Comments:

Reviewer #1:

Remarks to the Author:

The authors have done an excellent job addressing my concerns with the original manuscript submission. Great work! I applaud the authors for adding a thoughtful analysis of SARS-CoV-2 genome-wide crRNAs, experimentally testing the effect polyT included in their filtering strategy, repeating the tiled crRNA experiment, and testing additional crRNAs against the D614G. I believe this paper is ready for publication and would be a great addition to the Cas13 antiviral literature. However, given the authors made substantial changes to the manuscript text, I want to make a few minor suggestions:

1.Lines 89-92: Given that you cite Blanchard et. al. later in the text, the initial part of this statement claiming that Cas13 has not been shown to silence a replication-competent SARS-CoV-2 in mammalian cells is no longer true. Please update.

2.Line 101: Can you define here what is meant by predicted open regions of the Spike transcript? Does this mean stretches of non-structured RNA? Not bound by host proteins?

3.Line 151: Should the statement of predicted low off-target transcriptomic activity be reference Supplementary Table 5? If not, please explain why low off-target activity is predicted.

4.Line 179: spacer flanking sequences -> protospacer flanking sequences

5.Line 181 and 182: PSF -> PFS

6.Line 343: delete Cas13a in the reference. You may also want to note what Cas13 ortholog was used in the Blanchard et al. study since you mention the ortholog used in your own work.

7.Figure 1A and 2A: promoter details are included for PspCas13b and crRNA but not for the Spike-GFP and NCP-mCherry constructs.

8.A.U without the second period still present in Fig. 1, 2, and 3.

9.Maybe I was not clear in my initial response, but Fig. 1B and 2B images still in black and white which is confusing at initial glance, and having these black and white does not match the representation used in panels Fig. 1A and 2A. If possible, add color to Figure 1B and 2B to demonstrate GFP and mCherry signal in green and red, respectively.

Reviewer #2:

Remarks to the Author:

The authors have done an excellent job in addressing most of the queries made in the initial review.

A remaining concern for this study rests in the virological data as it was obtained from transfection of recombinant CRISPR gRNAs into infected virus target cells. As these transfections are limited to subpopulations of cells and may not parallel those infected with virus I had requested and believe still important that a single proof of concept experiment to be added with a relevant biological system such as lipid nanoparticles or with adenovirus vector delivery during natural infection.

The remaining responses were judged as completed and I have no further comments.

Manuscript No: NCOMMS-20-46711A

Dear Dr. Cloney,

We thank you for giving us the opportunity to revise our work, and the reviewers for their enthusiasm and constructive comments. We have revised our manuscript in response to the reviewers' comments and addressed all the editorial requests. Below, we offer a point by point response to each issue raised by the reviewers.

Reviewer #1 (Remarks to the Author):

The authors have done an excellent job addressing my concerns with the original manuscript submission. Great work! I applaud the authors for adding a thoughtful analysis of SARS-CoV-2 genome-wide crRNAs, experimentally testing the effect polyT included in their filtering strategy, repeating the tiled crRNA experiment, and testing additional crRNAs against the D614G. I believe this paper is ready for publication and would be a great addition to the Cas13 antiviral literature.

We thank the reviewer #1 for the positive feedback.

However, given the authors made substantial changes to the manuscript text, I want to make a few minor suggestions:

1.Lines 89-92: Given that you cite Blanchard et. al. later in the text, the initial part of this statement claiming that Cas13 has not been shown to silence a replication-competent SARS-CoV-2 in mammalian cells is no longer true. Please update.

We revised this statement to "however, the molecular basis by which Cas13 recognizes and suppresses replication-competent SARS-CoV-2 in infected mammalian cells, and its potential to suppress mutation-driven viral evolution and emerging variants remain to be established."

2.Line 101: Can you define here what is meant by predicted open regions of the Spike transcript? Does this mean stretches of non-structured RNA? Not bound by host proteins?

We changed "open regions" to "predicted non-structured RNA regions" as suggested.

3.Line 151: Should the statement of predicted low off-target transcriptomic activity be reference Supplementary Table 5? If not, please explain why low off-target activity is predicted.

At this stage of the manuscript, we do not yet discuss the off-targeting prediction in Suppl. Table 5. The prediction of low off-targeting is motivated by the required >21-nt basepairing between the spacer and target shown in Figure 1. We revised this statement for clarity to: "Taken together, these data indicated that crRNA2 requires >21-nt base-pairing with its target to trigger the necessary CRISPR-Cas conformational change and nuclease activation necessary for target degradation³⁰⁻³², and predicts low probability of off-target activity transcriptome-wide due to the extensive (>21-nt) spacer-target base-pairing required.

4.Line 179: spacer flanking sequences -> protospacer flanking sequences

We changed this to protospacer flanking sequences.

5.Line 181 and 182: PSF -> PFS

We changed this typo to PFS.

6.Line 343: delete Cas13a in the reference. You may also want to note what Cas13 ortholog was used in the Blanchard et al. study since you mention the ortholog used in your own work.

We revised this statement as suggested to "To the best of our knowledge, together with a recently published report by Blanchard et al⁴², these are the first two studies that definitively demonstrate the effective and specific targeting of replication-competent SARS-CoV-2 in infected mammalian cells using different CRISPR-Cas13 orthologs, and the first

demonstration of SARS-CoV-2 variants suppression with the CRISPR-pspCas13b ortholog.”

7. Figure 1A and 2A: promoter details are included for PspCas13b and crRNA but not for the Spike-GFP and NCP-mCherry constructs.

Promoter details are added in the figures as suggested.

8. A.U without the second period still present in Fig. 1, 2, and 3.

A.U revised to A.U.

9. Maybe I was not clear in my initial response, but Fig. 1B and 2B images still in black and white which is confusing at initial glance, and having these black and white does not match the representation used in panels Fig. 1A and 2A. If possible, add color to Figure 1B and 2B to demonstrate GFP and mCherry signal in green and red, respectively.

Fig 2B was changed from black and white to the original red color. We think conversion to black and white improves data presentation in Fig 1b due to the weak eGFP expression in VERO cells and the autofluorescence in the green channel generated with the medium resolution microscope used (EVOS M5000 FL Cell Imaging System). Representative color images from Fig 1b and Fig 2b are provided in the Source data file submitted with this manuscript with all normalized mean fluorescence values.

Reviewer #2 (Remarks to the Author):

The authors have done an excellent job in addressing most of the queries made in the initial review.

We thank the reviewer#2 for their positive and constructive feedback.

A remaining concern for this study rests in the virological data as it was obtained from transfection of recombinant CRISPR gRNAs into infected virus target cells. As these transfections are limited to subpopulations of cells and may not parallel those infected with virus I had requested and believe still important that a single proof of concept experiment to be added with a relevant biological system such as lipid nanoparticles or with adenovirus vector delivery during natural infection.

We thank the reviewer for this important point. We acknowledge that both lipid nanoparticles and adenoviral delivery strategies will allow better efficiency and therefore will bring this proof-of-concept one step closer to clinical translation. However, the development of these nanotechnologies and viral vectors will take a significant amount of time and delay the sharing of our findings with the scientific community. We will address this comment in a follow-up study and publish it in a separate manuscript.

The remaining responses were judged as completed and I have no further comments.